# Fire-induced rock spalling as a mechanism of weathering responsible for flared slope and inselberg development

Solomon Buckman [1✉], Rowena H. Morris[1] & Robert P. Bourman[1]

Inselbergs, such as Uluru in central Australia, are iconic landscape features of semi-arid and deeply denuded continental interiors. These islands of rock are commonly skirted by steep, overhanging cliffs (flared slopes) at ground level. The weathering processes responsible for formation of flared slopes and steep-sided inselbergs in flat, planated landscapes are enigmatic. One model emphasizes sub-surface weathering followed by denudation and excavation of saprolite to expose the unweathered bedrock while other models advocate slope development under subaerial conditions at ground level. We present a new hypothesis that identifies wildfire as a primary agent of flared slope development via fire-induced rock spalling around the periphery of inselbergs. Widespread fire-spalling following the 2019–2020 Australian fires illustrates that this is a common form of physical weathering in fire-prone environments but its effects are particularly evident in semi-arid regions where lateral fire-spalling dominates over fluvial and chemical weathering to create flared slopes and steep-sided inselbergs.

[1] GeoQuest Research Centre, School of Earth, Atmospheric and Life Sciences, University of Wollongong, Wollongong, Australia. ✉email: solomon@uow.edu.au

Ancient landscapes, such as central Australia, are characterised by deeply weathered bedrock and the development of well-preserved Cenozoic, multi-stage, and regolith profiles[1–3]. Isolated inselbergs, bornhardts, and mesas strewn across flat landscapes hint at a former higher relief[4–7]. Iconic examples of these steep sided inselbergs in Australia, include Uluru, Murphy Haystacks, Pildappa Rock, Burringurrah (Mount Augustus), and Katter Kich (Wave Rock)[4,5], which rise steeply from the surrounding landscape (Figs. 1 and 2) and in the case of Mt Augustus, may have been formed and exposed since at least the Jurassic[8]. The juncture between bare rock of the emergent inselberg and the surrounding plains covered by unconsolidated soil and sediment creates important sources of permanent water which promotes greater biodiversity in arid environments[9]. The peripheries of inselbergs are commonly the sites of rock shelters which represent important indigenous cultural sites that often host ancient rock art[10], which in Australia, provides a record of the world's oldest living civilisation[11,12]. Whilst the history of occupation of these rock shelters is the focus of numerous studies, the geomorphological mechanisms responsible for flared slope formation in different rock types are poorly understood.

The main agents of bedrock weathering, erosion and sediment production in mountainous continental regions such as Europe, Scandinavia and North America are generally attributed to flu-vioglacial processes or a combination of subaerial weathering and fluvial erosion in lower relief environments. Rates of vertical incision in active orogens vary between 1000 and 10,000 m Ma$^{-1}$ (1–10 mm yr$^{-1}$)[13]. However, flat, arid environments such as central Australia, are already denuded to local base levels across much of the planated landscape. This combined with very little rainfall, insignificant tectonic uplift, no recent volcanism and no glacial activity since the Permian or maybe the Cretaceous created flat landscapes that experience the slowest rates of erosion in the world[14]. Fission track studies[15–17] indicate relatively slow rates of 1–2 m Ma$^{-1}$ (0.001 mm yr$^{-1}$) of long-term landscape lowering throughout much of Australia during the Cenozoic. Cosmogenic studies reveal that climate is a major factor influencing rates of erosion with the lowest average rates of erosion of 1.5 m Ma$^{-1}$ occurring in arid Australia whilst the highest rates, of 35 m Ma$^{-1}$ are recorded from soil-mantled, spurs in humid temperate regions around the base of the SE escarpment[18]. Cosmogenic studies[14,19,20] of inselberg tops indicate rates of erosion of only ~0.3–0.6 m Ma$^{-1}$ (0.0003 mm yr$^{-1}$) while rates around the periphery of inselbergs are an order of magnitude faster (3–3.9 m Ma$^{-1}$) but highly variable[19]. Accordingly, the tops of inselbergs in Australia represent some of the most stable landscape features in the world. However, the development of steep, overhanging "flared slopes" around the periphery of the emergent rock hints at increased rates of weathering and the horizontal incision at or below ground level.

The term "flared slopes" is used to describe smooth, concave slopes at the junction between an emergent rock face and surrounding ground level[21] as seen at ground level around Uluru, Katter Kich, Murphy Haystacks, Pildappa Rock, Walga Rock and numerous other inselbergs in arid Australia (Figs. 1 and 2). Previously, these concavities have been interpreted to be the result of subsurface, moisture-generated, weathering in the scarp foot beneath saprolite, sediment, or soil cover[22]. After prolonged subsurface weathering, the soft saprolite or unconsolidated sediment is evacuated during periods of landscape lowering to leave a concave slope in the bedrock. However, others[2] point out the obvious relationship between the ground surface and flared slope and suggest that the current ground surface acts as a local base-level for the development of the overhanging slopes and only a thin veneer of soil covers the underlying, unweathered rock

platform surface. Hence, they suggest a subaerial origin for flared slopes based primarily on the observation that so many of the rock platforms are coincident with the modern land surface and lack any evidence of weathered saprolite beneath the thin soil around the margins of the inselberg.

The steep, near-vertical slopes around the periphery of inselbergs and the development of overhanging flared slopes at ground level (Figs. 2 and 3) indicate that differential weathering processes are operating more rapidly laterally, around the flanks of inselbergs than vertically on the tops[22,23]. Many of these overhangs host important indigenous rock-art, which are the focus of effective preservation methods to reduce erosion[10,24]. Flared slopes are well developed but not restricted to arid Australia. Overhanging flared slopes are common around the edges of prominent granite domes in the New England and Lachlan orogens of eastern Australia (Fig. 4) and at the base of the sandstone escarpments in the Sydney Basin (Fig. 5). These areas were heavily affected by the intense "Black Summer" fires of 2019–2020 as well as previous fires in 2013 (Fig. 5d). Here, we show the role fire plays in physically weathering exposed rock surfaces using examples following the Black Summer fires and discuss how this might be responsible for flared slope development.

## Results
Establishing the variables involved in rock weathering and fire behaviour is a key aspect of developing an accurate fire-induced rock spalling hypothesis. We expand on these variables by drawing on field observations and existing findings outlined below.

**Mechanical weathering.** The physical breakup and removal of rocks of varying hardness and degrees of weathering via mechanical weathering is the primary process that denudes and sculpts uplifted regions of Earth's surface. Sub-critical cracking describes the slow propagation of microfractures through a rock in low-stress, near-surface conditions as a result of thermal stress, ice wedging, mineral alteration (volumetric expansion) and bio-mechanical processes such as root growth[25]. Sheeting is characterised by thick (0.1–1 m) layers of rock peeling off exposed surfaces roughly parallel to the surface topography. There is debate as to whether sheeting is related to gradual unloading and release of stresses near the surface or a combination of other stresses[26,27]. The physical process of thermal expansion and contraction of rocks over thousands of years is responsible for the thinner, gradual flaking (exfoliation) of rock surfaces, which can be observed all over the surface of inselbergs in central Australia[5] and presumably the main process responsible for the slow rates of erosion at the tops of inselbergs[14].

Fracture propagation is facilitated by the presence of water[28], which helps to break chemical bonds leading to more fractured rock at shallow, superficial levels of the crust. Thus, rocks are generally more fractured in the superficial, near-surface environments than at deeper levels. Spontaneous rock-burst events were captured on video during a hot summer of 2014 in California when a granite dome at Twain Harte began explosively exfoliating[29]. Extreme thermal stresses associated with fire and lightning strikes are acknowledged as mechanisms of critical stress fracturing in rocks but generally considered to be a rare form of rapid and catastrophic mechanical weathering[25]. Our observations of rock surfaces following wildfires are that fire-related rock spalling is a commonly observed phenomenon wherever high-intensity fire has swept across rocky outcrops (for example, Figs. 5 and 6). We suggest that fire-spalling is a significant driving mechanism of physical weathering in arid,

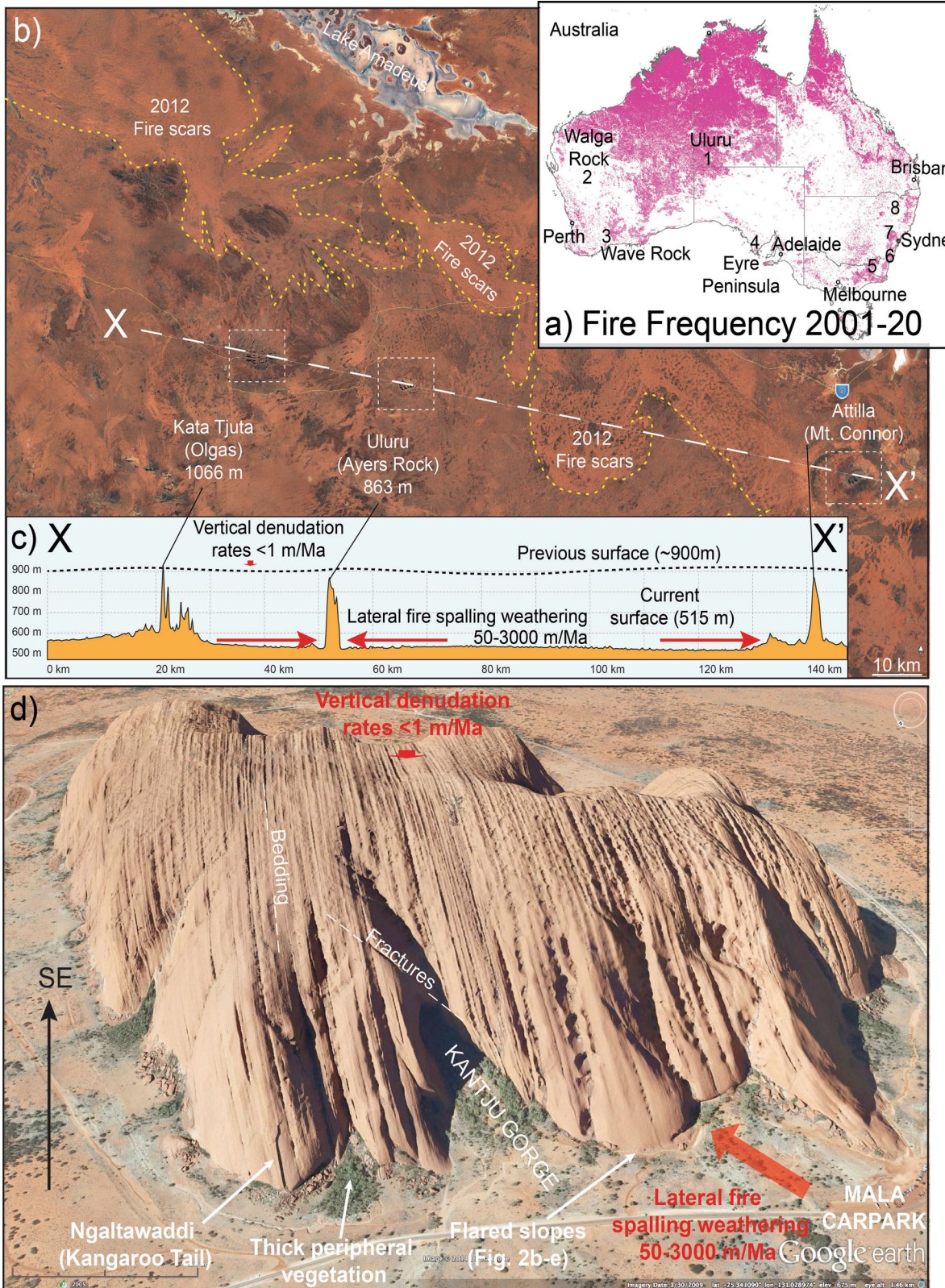

**Fig. 1 Study site setting. a** Location of study sites and the distribution and frequency of fires in Australia since 2001 sourced from NASA's Fire Information for Resource Management System (FIRMS) (https://firms.modaps.eosdis.nasa.gov/map/#d:2020-07-29..2020-07-30;@0.0,0.0,3z). Greater intensity of the pink colour indicates increased fire frequency. Australia has been widely affected by fire over the past 20 years and that the grassland savannah regions of northern Australia experience regular, almost annual rates of fire recurrence, whilst the forests of SE Australia experience less regular but more intense fire regimes. **b** Aerial view of Uluru region showing recent fire scars and location of topographic profile X-X'. **c** Topographic profile X-X' through Kata Tjuta, Uluru and Mount Connor. **d** Aerial perspective view of the iconic inselberg Uluru (Ayres Rock) in central Australia and the location of significant flared slopes around its periphery. Source: Google Earth.

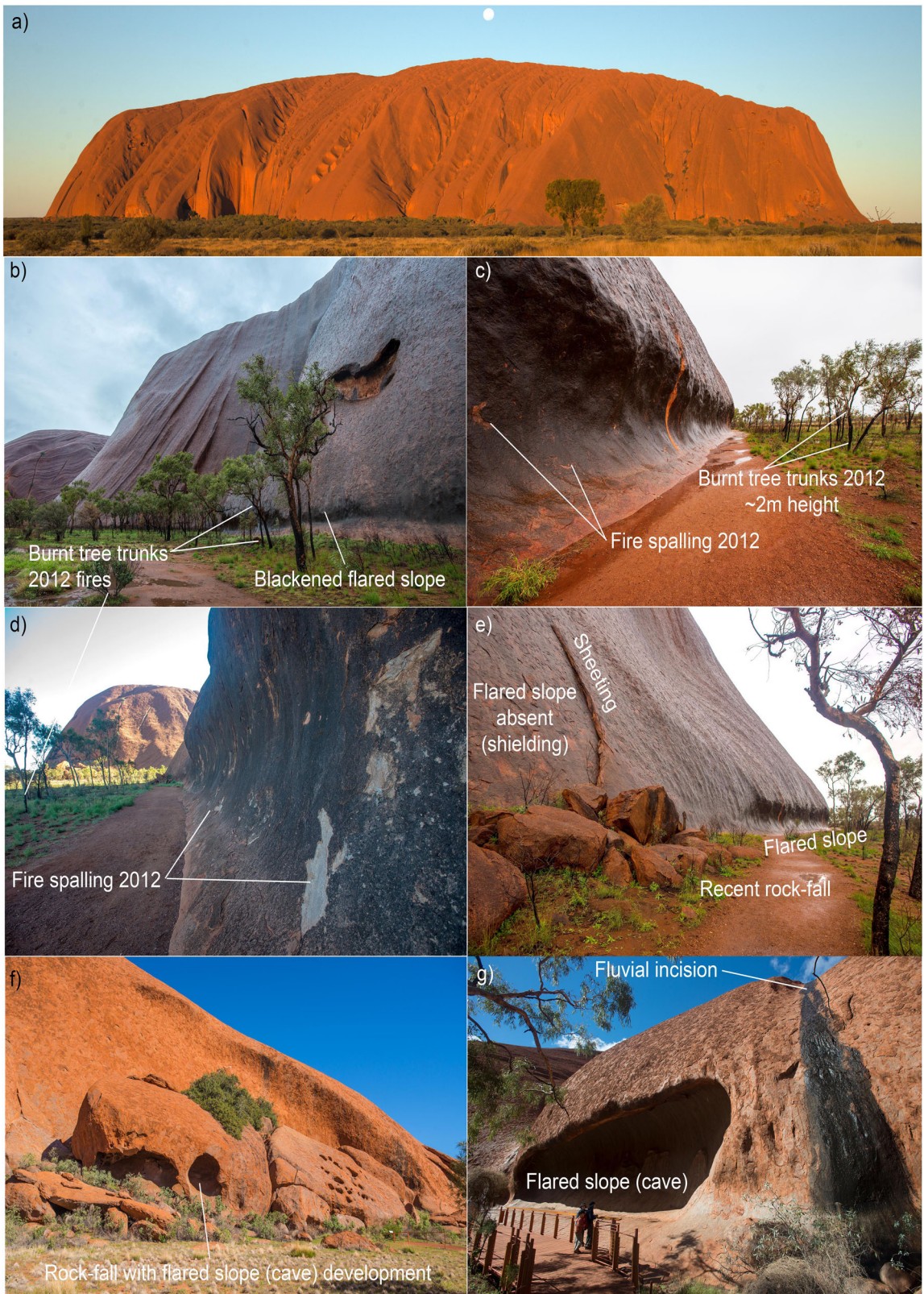

fire-prone environments and has been overlooked as an important agent of geomorphic change and landscape evolution.

**Wildfire temperatures**. A detailed study of high-intensity wildfires in eucalypt forests of SW Australia[30] revealed that these fires burn at temperatures between 300 °C at the tips of visible flames and up to a maximum of 1100 °C near the flame base, while temperatures of up to 1330 °C were recorded in Canadian crown fires[31]. Experimental fires conducted in jarrah forests of southwest Western Australia (Project VESTA) reveal that temperature correlates directly with the rate of spread, fire intensity, flame height and surface fuel bulk density[30]. This single case study measured the average flame-front residence time in eucalypt forest fuels of about 37 s. However, radiant heat and hot winds

**Fig. 2 Flared slopes at ground level around Uluru. a** The steep-sided inselberg of Uluru in central Australia. **b**, **c** Flared slope development at the foot of Uluru produces a remarkably uniform concave surface roughly 2 m high coincident with the level of the surrounding plain. Note the recently burnt trees and the height of the charcoaled trunks, which matches the height of the flared slopes, which themselves are blackened from this recent fire. **d** Fire-spalling on the arkose sandstone rock surface caused by a fire in 2012. **e** Sheeting and rockfall events produce large boulders at the foot of the inselberg which shield the inselberg from further fire-spalling and hence the lack of concave flared slope development behind the fallen boulders. **f** Rockfall boulders have themselves developed small caves on the side exposed to the vegetation and approaching fires indicating flared slope development occurred sub-aerially after the rockfall event. **g** Mala Puta Cave displays extensive flared slope and cave development at ground level. Mala Puta Cave is 2–3 m deep. In this example, the wet, black waterfall on the right highlights the difference in rates of fluvial incision compared to the rate of lateral erosion due to flared slope development.

fanning out in front of the fire have the ability to pre-heat the rock surface and vegetation before and after the arrival of the fire front[31] particularly along cliff lines.

We report the first documented case of spalling in basalt from Mount Kaputar in northern N.S.W. (Fig. 4d). Basalt is a high-temperature volcanic rock with no quartz content. Fire-spalling was minimal across most of the outcrops and generally consisted of dislodged pyroxene phenocrysts. However, a few basalt outcrops adjacent to nearby fallen burnt logs were intensely scorched and displayed thin (1–4 mm) spalled flakes of basalt indicating that fire-spalling is not restricted entirely to quartz-rich lithologies. In mature eucalypt forests with large, woody fuels, termed 'down wood'[32], fires can burn or smoulder for days, providing prolonged heat required for extensive spalling. Some cliff faces record distinct ghosted impressions of nearby tree trunks with the resultant spalling hollowing out the line and shape of a tree trunk in an otherwise flat, vertical rockface (Fig. 5a —right-hand side). A discarded brown glass bottle adjacent to the basalt spalling had softened and undergone ductile collapse and partially melted. The glass had cooled slowly enough to avoid shattering indicating prolonged heating from the smouldering downward. This glass was collected and placed in a high-temperature oven where it was observed to become soft and malleable at 750 °C and completely collapsed and started melting at 830 °C indicating that this fire sustained ground surface temperatures of between 750 and 830 °C next to the smouldering tree and fallen logs.

**Fire-induced rock spalling.** Fire is known to accelerate the rock flaking process[25,33–38] resulting in rock spalling[36,39] and shattering[38]. Conflagration leads to the rapid disintegration of the rock surface due to the differential expansion of the hot rock surface compared with the cooler interior. Fire-spalling can remove between 10 and 100% of the burnt rock surface in sheets between 5 and 50 mm thick[37] depending upon rock type and fire intensity. Detailed measurements of post-fire rock spalling after the Esperanza chaparral fire in California revealed that 7–55% of the granodiorite boulder surfaces were spalled to a depth of 11–24 mm[33]. They found that the thickest spalled sheets occurred around the flanks of the boulders and cautioned that, if sampled for cosmogenic dating, these freshly exposed, spalled surfaces would produce a significant underestimate of exposure age. These figures match our own observations of spalled granite following the fires in Cobargo, Moonbi and Thredbo N.S.W. (Fig. 4) in which granite boulders spalled sheets between 5 and 50 mm thick, while sandstones from the Blue Mountains spalled sheets between 5 and 22 mm thick (Fig. 5).

Quartz expands four times more than feldspar and twice as much as hornblende and shows a 3.76% volume expansion when heated from room temperature to 570 °C[40]. Thus, quartz-rich rocks have a greater expansion potential and are more likely to spall. Experimental studies[41] show that rock elasticity reduces significantly at temperatures as low as 200 °C, over a relatively short period of time. Goudie et al.[41] postulated that rock outcrops

subject to intense fires would have an increased susceptibility to erosion via spalling and weathering. However, these findings have not been applied to broader landscape models or the formation of flared slopes around inselbergs.

**Fire regimes.** The potential rate of erosion due to fire-spalling at the base of inselbergs will be strongly influenced by fire severity and recurrence intervals, which vary greatly across Australia from 1- to 5-year recurrence intervals and <10,000 kW m$^{-1}$ for tropical savannas of the north, to > 100-year intervals and >10,000 kW m$^{-1}$ for tall, open forests of the cool, temperate south[42]. Accurately calculating the fire return period is difficult due to limited historical records but estimates for arid, spinifex-dominated regions such as the Tanami are in the order of every 7–9 years[43]. Analyses of satellite data between 1998 and 2004 revealed that 27% of arid Australia burnt at least once over that 6-year period[44]. Figure 1a shows the areas burnt in Australia since 2001. The surface area of the rock affected by spalling depends on the rock-type and severity of the fire. Fire severity is strongly determined by the bulk density[45], height and proximity of the adjacent vegetation to rock surfaces and the surrounding slope gradient. All the examples of flared slopes shown in Figs. 1 and 2 reveal a close relationship between the height of the encroaching vegetation and the height of the concavity. Katter Kich, Pildappa Rock and Walga Rock form distinct embayments where the flared slopes are most pronounced, which appear to promote denser, taller vegetation growth and hence greater fuel loading and thus higher fire severity (Fig. 3).

The impermeable nature of inselbergs results in rapid and efficient water runoff from the bare-rock surface before draining into adjacent, thin soil profiles. This creates a "roof and gutter" effect around the periphery of many inselbergs which creates permanent water holes and shallow groundwater within easy reach of deep-rooted plants. Inselbergs create important geodiversity within otherwise flat landscapes and thus host important niche ecosystems that add to the overall biodiversity of desert regions[46]. Accessible groundwater around the fringes of the inselbergs encourages denser, taller vegetation at the interface between bare rock and unconsolidated surficial sediments which in turn increases the fuel load. Inselbergs are prominent topographic features in flat deserts that provide sources of permanent water, abundant flora and fauna and shelter.

Grassy plains and savannahs of central Australia are characterised by regular, low-intensity fires with fire recurrence intervals between 1 and 5 years[42]. However, where these fires encounter inselbergs they move into thicker, taller vegetation regimes with greater fuel loads (Figs. 2 and 3). Inselbergs are topographic highs within relatively flat landscapes and the slight increase in slope gradient around the inselberg will accelerate and intensify an approaching fire front. Steep slopes around the margins of inselbergs possibly act as chimneys, drawing in hot air from the surrounding plains and channelling them upwards. These factors possibly help to draw in fires from the surrounding

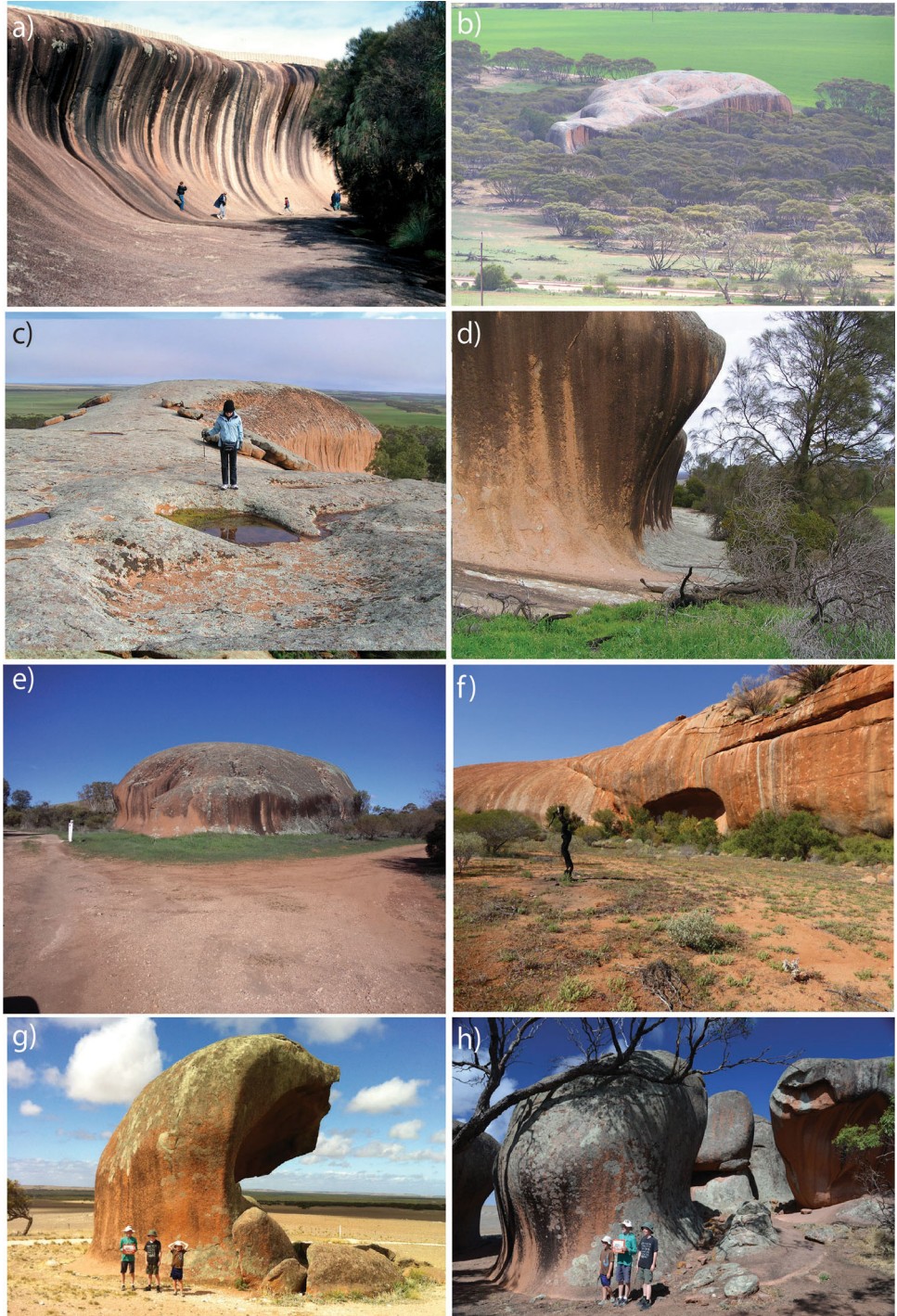

**Fig. 3 Flared slopes around the periphery of prominent inselbergs. a** A 15 m high flared slope at Katter Kich (Wave Rock), Hyden, Western Australia. A thin wedge of soil overlies fresh, unweathered granite and becomes progressively deeper away from the inselberg enabling the growth of dense, flammable vegetation, which encroaches on the inselberg. **b** Flared slopes developed in granite at Turtle Rock, Eyre Peninsula, South Australia, surrounded by dense vegetation. The photo was taken from atop Mount Wudinna looking NW. **c** The top of Pildappa Rock, Eyre Peninsula, South Australia is covered in old, lichen growths whilst the flanks are relatively devoid of any lichens. **d, e** Well-developed flared slope on the margin of Pildappa Rock showing encroaching vegetation growing on a thin veneer of soil. **f** Walga Rock in Western Australia showing dense vegetation around the periphery of the inselberg due to the "roof and gutter" effect of the emergent, impermeable granite inselberg. **g, h** Murphy Haystacks in South Australia displaying flared slopes at ground level on all sides of the granite inselberg. Overhanging caves develop on the right side due to the prevailing south-westerly winds in this region, which would control the direction from which most fire fronts would approach the rocks.

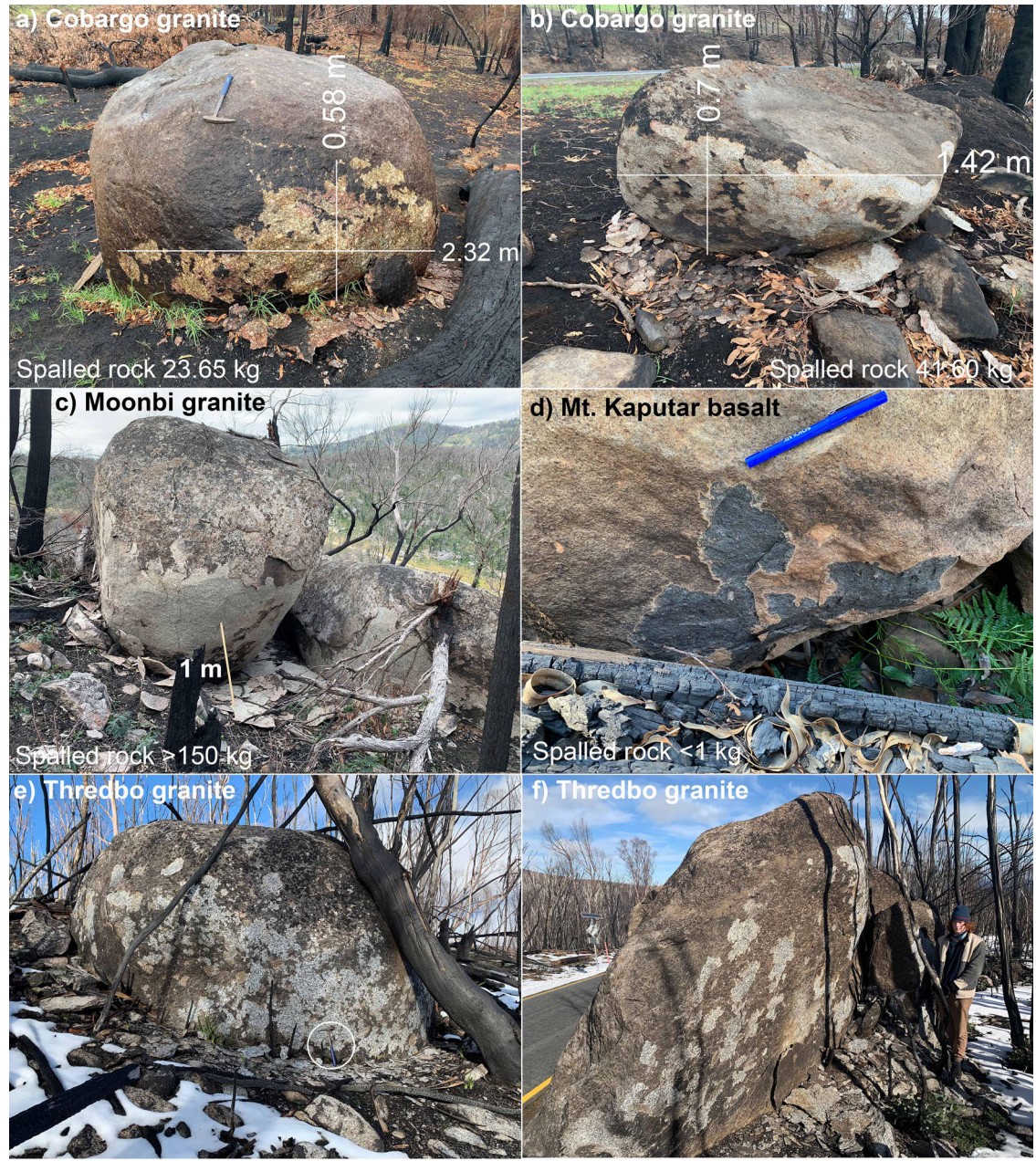

**Fig. 4 Fire-related spalling on granite and basalt boulders of eastern Australia following the 2019–1020 fires. a** Granite tor near Quaama spalled 23.65 kg of rock on the facing surface due to the intense and prolonged heat provided by the adjacent burnt log. Location −36.509178, 149.831598. **b** Spalled granite boulder near Cobargo yielded 41.60 kg of spalled rock of which 12 kg was attributed to a single spalled slab visible on the lower right of the boulder. Location −36.404498, 149.931435. **c** Intensely spalled Moonbi Granite near Tamworth, N.S.W., showing not one but several generations of spalled sheets flaking off at least 150 kg of rock during a single, intense fire in 2020. Location −30.942304, 151.108704. **d** Thin (2–4 mm) spalled flakes on a Cenozoic basalt outcrop at Mount Kaputa, northern N.S.W., and the location of the melted glass bottle. Location −30.308181, 150.210553. **e, f** Spalled granite boulder at Thredbo, N.S.W. in Australia's most elevated, alpine environment in the Snowy Mountains. Rock pick (42 cm) (circled) and person (190 cm) for scale. Location −36.533969, 148.233443.

plains into and around topographically high inselbergs where the intensity is enhanced at the base of inselbergs due to the denser vegetation and greater fuel load.

**A fire-induced spalling weathering formula**. Fire-spalling leads to physical weathering (erosion) and disintegration of exposed rock faces[36,37] as shown in Figs. 3–5. The degree and extent of spalling on different rock types and at varying temperatures and durations is less well understood and requires further

experimental work[41] but essentially fire-spalling is a function of fire intensity (temperature), duration and rock type with quartz-rich rocks having a greater propensity to expand and spall[40].

We developed a simple fire-spalling erosion formula to estimate a long-term rate of fire-induced spalling that broadly considers the net result of fire-spalling in terms of the thickness (width) of the spalled flakes produced by a single fire event, the total surface area as a percentage of the exposed rock face affected by a single fire-spalling event, and the average fire recurrence interval for a given region. Together these variables can give some

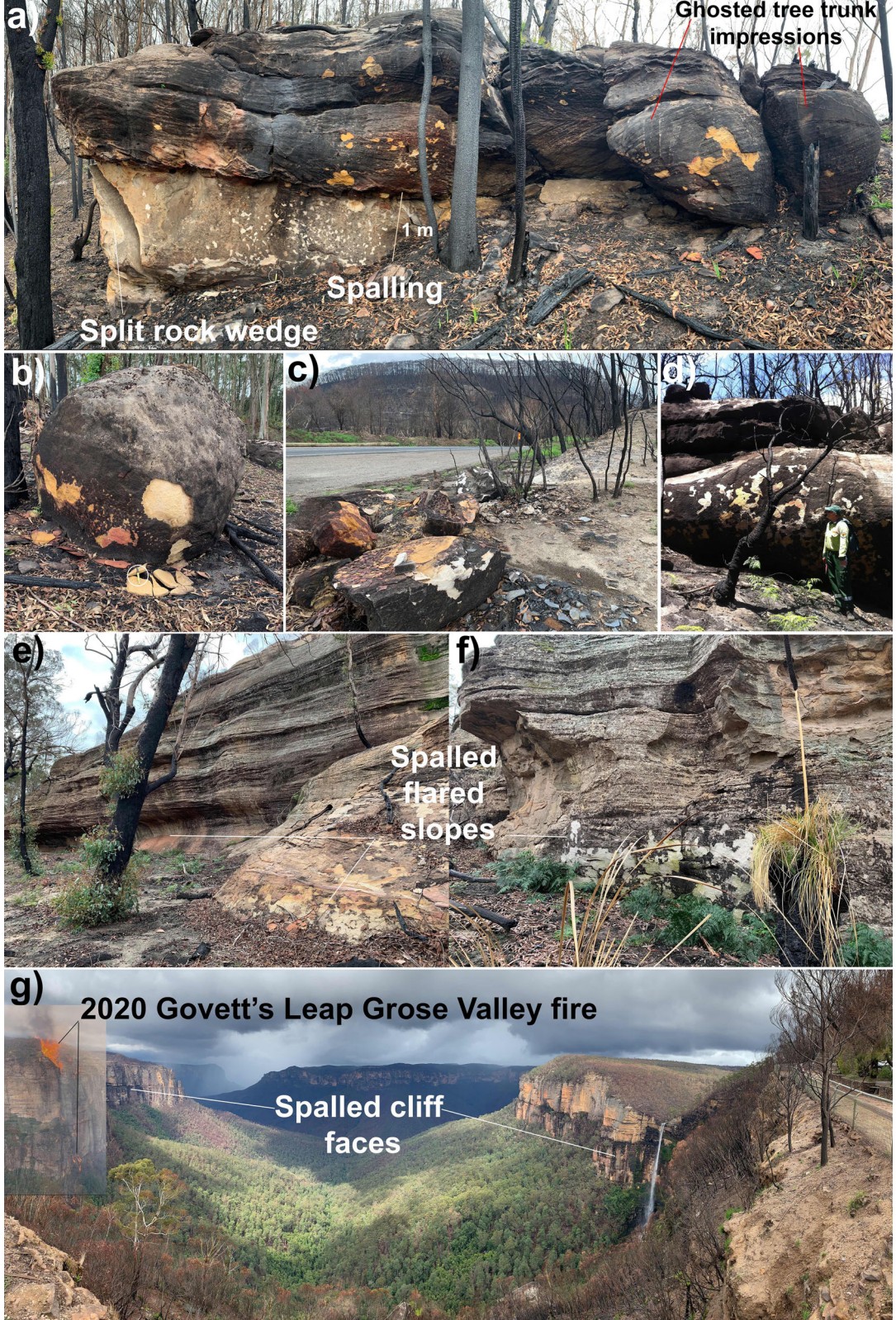

indication as to the long-term rates of erosion due to fire-spalling at the base of an inselberg or cliff face where there is significant vegetation to fuel a wildfire.

The formula for erosion due to fire-spalling.

$$E = \frac{W \times A}{t} \qquad (1)$$

where, $E$ = rate of erosion due to fire-spalling (mm yr$^{-1}$), $W$ = average width (thickness) of spalled sheets (mm) for a single fire event. Dependent on rock type (quartz content and texture), rock strength, fire temperature and duration, $A$ = area of rock surface affected by fire-spalling as a percentage (%) of total surface area. Dependent on temperature and duration of the fire, $t$ = average fire recurrence interval (years). Determined from regional,

**Fig. 5 The effects of fire-spalling on quartz sandstone (Hawkesbury Sandstone) of the Sydney Basin following the 2019–2020 fires of eastern Australia. a** Spalled yellow cross-bedded sandstone overlying white, spalled massive sandstone at Wynnes Rock Lookout, Mount Wilson showing 'ghosted' impressions of burnt tree trunks adjacent to the rock face. Location −33.521348, 150.370183. **b** Sandstone boulder shedding yellow spalled sheets 2.2 cm thick. **c** Spalled sandstone blocks on the side of the road with the burnt Mount Tomah in the background. Location −33.549054, 150.396409. **d** Nepean fires of 2013 produced spalling on sandstone—note the intensity of spalling is greatest adjacent the completely burnt tree hanging over the rock face. Location −34.329362, 150.636456. **e, f** Flared slopes at the base of quartzose sandstone cliffs at Rylstone in the western Sydney Basin. **g** The Gross Valley from Evan's Lookout at Blackheath shows the fire scars from flames that burnt up the 200 m cliff face and onto the plateau as shown by the superimposed image of the 2019–2020 fire on the left. Fire-spalling is evident on all of the blackened cliff faces.

historic fire records or palaeofire records for longer time periods. Dependent on vegetation, climatic regimes and land management practices.

*Limitations*: this equation applies to a near-vertical rock face at ground level which receives uniform heat radiation from a fire that burns right up to the rock face at ground level. The intensity of radiation will vary according to the dynamics of the fire front, fuel loading, vegetation type and slope gradient. Flame height is not critical to the overall rate of retreat of the cliff face because fire-spalling at the base of the cliff will gradually remove material supporting the cliff resulting in over-steepening at the base of the cliff and periodic sheeting and rockfalls as the overhanging cliff face becomes gravitationally unstable. The formula assumes that fire recurrence intervals have remained constant but we know from palaeofire records[47,48] that fire intensity and recurrence intervals are largely controlled by long-term climatic variations which affect vegetation types and thus fuel loads. Below, we give two end-member examples of long-term rates of spalling-related erosion for low and high-frequency fire regimes that may apply to temperate and arid environments, respectively.

Example 1. Low intensity, irregular fire regime. In this scenario, the average fire against a cliff results in spalling and flaking of ~10 mm sheets off ~20% of the surface area at ground level during a single fire event. Fire recurrence interval is one event every 50 years.

$$E = \frac{W \times A}{t} = \frac{10\,\text{mm} \times 0.2}{50} = 0.04\,\text{mm yr}^{-1} = 40\,\text{m Ma}^{-1}$$

Example 2. High intensity, high-frequency fire regime. In this scenario, the average fire against a cliff results in spalling and flaking of ~20 mm sheets (Fig. 5) off ~80% of the surface area at ground level. Fire recurrence interval is one event every 5 years.

$$E = \frac{W \times A}{t} = \frac{20\,\text{mm} \times 0.8}{5} = 3.2\,\text{mm yr}^{-1} = 3200\,\text{m Ma}^{-1}$$

In an intensely fire-prone environment such as example 2 above, it may only take about 625 years of fire-induced spalling to weather out a 2 m deep flared slope at the base of a vertical rock face. The point at which undercutting due to fire-spalling would trigger massive sheeting of the unsupported, overhanging rock ledge and subsequent rockfall event is not well constrained but some of the flared slopes around Uluru and Walga Rock are at least 2–3 m deep (Fig. 2h).

**Sediment production rates**. If rates of erosion due to fire-spalling around the periphery of an inselberg are orders of magnitude greater than those across the top of the inselberg, then this has implications for mechanisms of sediment production in flat, arid environments like Central Australia.

Spalling of a 20 mm sheet from a 1 m² area of granite with a density of 2691 kg m⁻³ will yield 0.02 m³ (53.82 kg) of rock. A flared slope around an inselberg such as Uluru with a circumference of ~10,000 m and a height of 2 m, would produce 400 m³ (1,076,400 kg) in a single event in which 100% of the 2 m high flared slope was spalled. Obviously, 100% spalling of the

entire flared slope would never occur in a single event, so we use the long-term erosion rate based on fire recurrence intervals and average area spalled calculated in Eq. 1. This long-term estimate of sediment production from a single inselberg is compared with quantitative measurements of spalled granite surfaces sampled after the 2019–2020 fires in Cobargo on the south coast of N.S.W., Australia.

The formula for sediment production.

$$S_{\text{FS}} = P.H.E$$

where, $S_{\text{FS}}$ = sediment production from fire-spalled rock surface (cubic metres per year), $P$ = perimeter of the inselberg (metres), $H$ = height of the flared slope around the inselberg as determined by vegetation and fire height, $E$ = rate of erosion due to fire-spalling (Eq. 1).

Fire-spalling sediment production around the periphery of an inselberg such as Uluru with a perimeter of roughly 10,000 m and flared slope height of 2 m, would be

$$S_{\text{FS}} = 10,000\,\text{m} \times 2\,\text{m} \times 0.0032\,\text{m yr}^{-1} = 64\,\text{m}^3\,\text{yr}^{-1} = 172,224\,\text{kg yr}^{-1}$$

This can be standardised to give a volume of rock spalled per year per square metre, which is the same as the erosion rate but in cubic metres per year. Given the density of the rock (granite = 2691 kg m⁻³ and compacted, meta-arkose sandstone (Uluru) are about the same) we can calculate the average mass of rock spalled each year. In the above scenario, it equals 8.61 kg per square metre per year.

The rate of background (non-fire related) sediment production ($S_{\text{BA}}$) from erosion of the surface area of an inselberg such as Uluru is equivalent to the surface area (~3,440,000 m²) multiplied by the average denudation rate of ~0.3–0.6 m/Ma (0.0003 mm yr⁻¹) as established from cosmogenic studies.

$$S_{\text{BA}} = 3,440,000\,\text{m}^2 \times 0.0000003\,\text{m yr}^{-1} = \sim 1\,\text{m}^3\,\text{yr}^{-1} = 2691\,\text{kg yr}^{-1}$$

This equates to only 0.00081 kg per square metre per year. We estimate that fire-spalling on a 2 m high perimeter produces in the order of 64 times more sediment than the erosion of the entire surface of the inselberg due to background (non-fire related) processes.

Spalled granite material was collected from two locations following the 2019–2020 fires in the Cobargo region along the south coast of N.S.W. (Fig. 4) to assist in quantifying the amount of rock spalled from a single rock face. Spalled surface area can be estimated simply by measuring the maximum height and width of the spalled surface in the field. We also created a digital surface using photogrammetry MetaShapePro software to calculate a precise surface area of the spalled surface. All of the spalled material was weighed and a standard granite density of 2691 kg m⁻³ was used to determine total volume. Generally spalling occurs as thin (1–3 cm) sheets but occasionally includes large 20–30 cm thick slabs that substantially add to the overall weight of spalled material. Whilst complete spalling of a 2 cm sheet from one square metre of granite surface will produce 53.82 kg m⁻² of rock, our two sites (Cobargo2 and 3A) produced 23.65 kg total (16.89 kg m⁻²) and 41.60 kg m⁻² total (33.55 kg m⁻²), respectively, indicating an

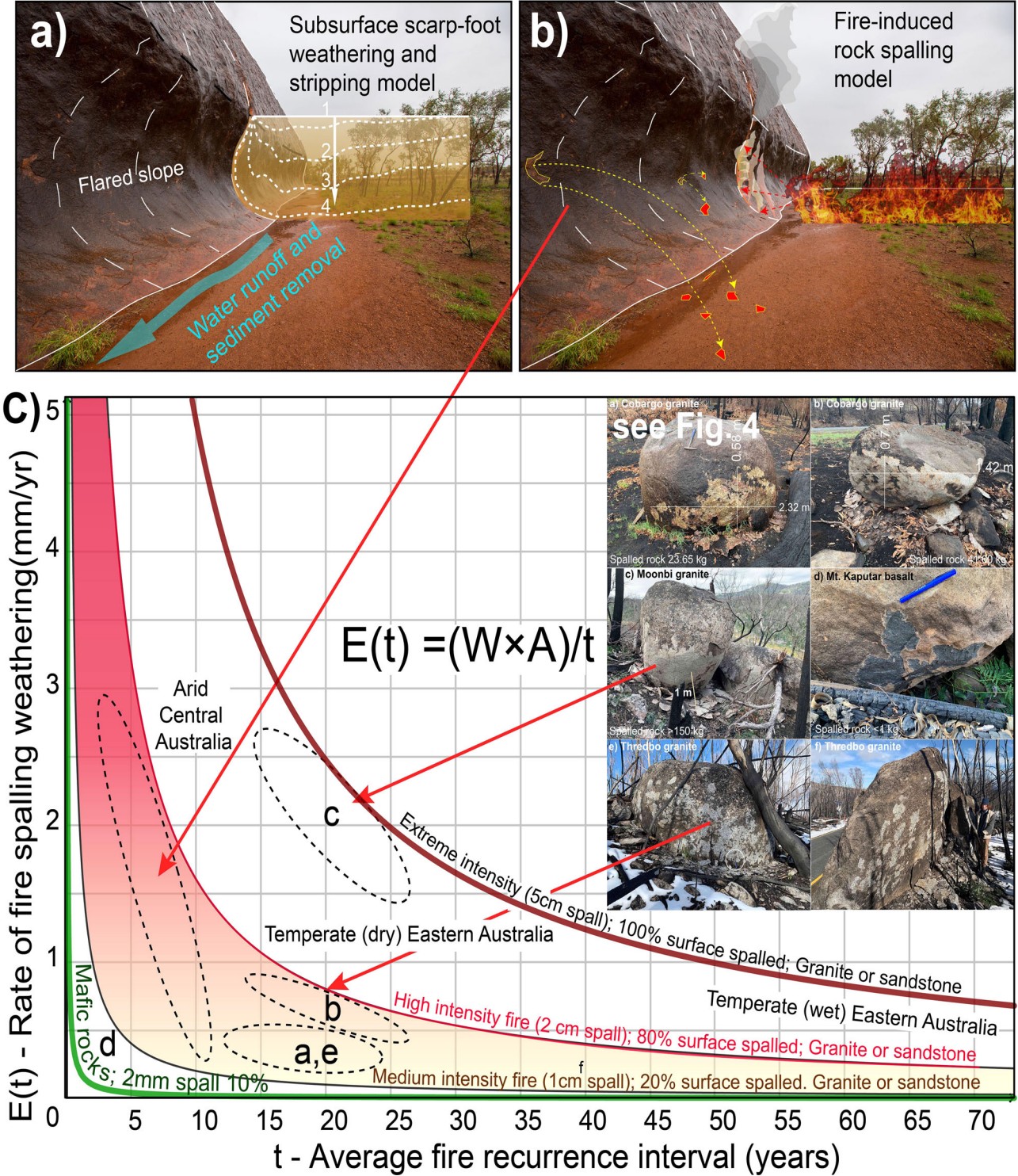

**Fig. 6 Existing and new models of flared slope development using Uluru as an example and a graphical representation of the formula for erosion due to fire-spalling in different fire regimes. a** The model of Twidale and Bourne 1998[22] involving subsurface weathering via shallow groundwaters to form soft regolith or unconsolidated soil that is subsequently removed by erosion and landscape lowering; **b** a new model of flared slope development via fire-induced rock spalling associated with episodic wildfire events. Note the charcoal on the recently burnt trees is the same height as the flared slope; **c** inverse correlation relationship between rates of erosion E(t) plotted against fire recurrence interval (t) using the formula E = W.A/t (see https://www.geogebra.org/calculator/uwa68amr). Rock-type and fire temperatures tend to control the thickness of spalled sheets (W) whilst fire intensity and duration are the main controls on the surface area spalled (0–100%). The inverse correlation of rates of fire-spalling erosion with average recurrence intervals (t) results in an increasing rate of weathering with smaller average fire recurrence intervals. Fire recurrence intervals are largely controlled by climatic and vegetation regimes and examples from Figs. 2 and 4 are shown and plotted on the graph according to the fire recurrence interval for that region.

average spalling thickness of 0.63–1.25 cm although the spalled thickness was highly variable with spalling distinctly more prominent along sharp or protruding edges than on flat surfaces. Large logs or tree trunks have the potential to continue burning long after the fire front moves through and their presence near rock surfaces significantly increases the degree of spalling. The most intense spalling was observed at Moonbi Granite near Tamworth in northern N.S.W. where some granite boulders had 100% surface spalling up to 2 m above ground level and not one but several spalled sheets (5–20 cm total thickness) exfoliating off during a single, intense fire creating several hundred kilograms of spalled rock debris on the granite surface facing the fire front (Fig. 4). Likewise, lichen coated granites from Australia's most elevated alpine regions in the Snowy Mountains (Thredbo) displayed intense spalling but were covered in snow six months later. The fire recurrence interval for these alpine regions is probably in the order of one every 20–100 years thus the effects of fire-spalling are less pronounced than in arid regions and less evident than other forms of fluvial or chemical weathering that dominate in wetter climates. However, the abundant spalled surfaces shown in Fig. 4 reveal that large, intense fires such as the Black Summer fires of 2019–2020 will result in significant erosion and sediment production even in alpine environments.

## Discussion

The development of vertical to overhanging flared slopes at ground level around the peripheries of inselbergs is direct observational evidence of the contrasting rates of vertical erosion operating slowly on the tops of inselbergs but much faster, lateral erosion at the base and around the periphery. We propose that exposure of rock surfaces to frequent, high-intensity wildfires over millennia creates a differential rate of lateral erosion at ground level around the periphery of inselbergs compared to slower vertical rates of erosion operating at the tops of the inselbergs. In arid areas where fluvial and chemical weathering processes are relatively slow-acting, fire-spalling results in the development of flared slopes and prominent inselbergs. Higher fire frequencies and intensities should accelerate lateral erosion at ground level, particularly along the thickly vegetated bases of inselbergs and escarpments. Even low-intensity fires such as the 2012 fire around Uluru resulted in minor spalling on flared slopes that were buffered from adjacent vegetation by a 2–5 m cleared walking path (Fig. 2). Fire-induced rock spalling and flared slope development undermines and potentially destabilises the outer shell of the inselberg at ground level, occasionally resulting in large-scale sheeting (~1–2 m thick sheets) and rock-fall events. Paradoxically, rockfall debris has the effect of temporarily shielding the inselberg from subsequent wildfire events resulting in no flared slope development on the inselberg surface behind the boulders, although the fallen boulders themselves are often subject to flared slope development on their exposed outer surface (Fig. 2e, f). We suggest that wildfire is largely responsible for the initial disintegration of rock in fire-prone environments and is an important process in the physical breakdown of rock and production of sediment, particularly in dry, flat continents such as Australia. We provide a simple fire-spalling erosion formula to predict the rates of fire-induced rock spalling for different fire regimes that can be applied to broader landscape models.

An inselberg such as Uluru has outcrop dimensions of roughly 2.5 × 1.5 km. The fire-spalling erosion Eq. 1 indicates that an intense fire regime with a fire recurrence interval of 5 years (Example 2), could result in a maximum, long-term, lateral erosion rates of about 3 mm yr$^{-1}$. Assuming that fire-spalling operates on all sides of the inselberg, this gives a combined rate of 6 mm yr$^{-1}$ total. At this rate, the narrowest section of an inselberg the size of Uluru (1,500,000 mm) would be completely removed from the landscape in about 250,000 years. In reality, the rate of retreat would be much slower when we consider the shielding effect that rockfalls have in terms of temporarily protecting the inselberg and slowing the overall rates of rock spalling (Fig. 2g). Conversely, projected backwards, the periphery of an inselberg like Uluru may have been up to 200 m wider when humans first arrived in Australia some 65,000 years ago[49]. Twenty million years ago, Uluru was probably at the centre of a vast, undissected plateau stretching 120 km across from north to south and connected to other remnants of this ancient plateau, for example, Kata Tjuta (the Olgas) to the west and Attila (Mount Connor) to the east (Fig. 1).

We present a graph (Fig. 6c) that plots the rates of erosion due to fire-spalling ($E_{(t)}$) vs. fire recurrence intervals (t) to illustrate the formula in Eq. 1. Rock type and fire intensity control the thickness of spalled sheets with quartz-rich rocks such as granite and quartz sandstones displaying the most intense degrees of spalling (2–10 cm thick). In contrast, rocks such as basalt rarely spall and if so the spalled sheets are only a few millimetres thick and only occur as small patches and only in the most intense heat-affected rockfaces. Weathering of mafic rocks such as basalt is probably controlled primarily by the chemical breakdown of unstable, high-temperature minerals such as olivine, pyroxene and plagioclase rather than the physical process of spalling. The degree of spalling on a rock face is measured as a percentage of the total surface area and is controlled mostly by fire intensity and duration. The Moonbi Granite in Fig. 3 showed evidence of 3–4 sheets each about 2 cm thick being spalled off during one intense fire resulting in 100% surface spalling with a thickness of 5–20 cm. This was the most intensely spalled example we have encountered but given we were restricted to roadside outcrops we suspect there are more intense examples of spalling to be discovered. Whilst the relationships between rates of erosion with the thickness ($W$) and area ($A$) are linear, there is an inverse correlation with fire frequency ($t$) that creates a non-linear hyperbolic relationship (Fig. 6c). Dry, arid regions with short fire recurrence intervals of only 5–10 years (left side of the graph) will potentially experience increasingly higher rates of fire-spalling erosion than wet temperate regions with fire recurrence intervals of >50 years (right side). Examples of fire-spalling are added to provide context to the different rock types and climatic regimes across Australia.

In stark contrast, the rates of background, fluvial-related and/or flaking-related erosion on top of an inselberg are an order of magnitude lower at about 0.3 m Ma$^{-1}$ [14]. At these slow rates, Uluru which sits 348 m above the surrounding plain, would take in the order of 1.16 billion years to be completely planated. This rate is obviously much slower than the overall rates of denudation for these regions and does not reflect the rate of scarp retreat and inselberg formation observed around rocks that are themselves are only a few hundreds of millions of years old, as for example, the heavily spalled Permian–Triassic granites in the New England Orogen of eastern Australia (Fig. 4).

Whilst fire-spalling was identified as a mechanism of physical weathering as early as 1927[36] it was largely regarded as an isolated, local phenomena. We recommend that fire-spalling needs to be incorporated into erosion and landscape evolution models that currently tend to focus almost entirely on fluvial, glacial, periglacial and erosive mass wasting processes operating within drainage basin catchments[50]. Wildfires are high-energy, episodic events that disintegrate rock and present a massive departure from the slow steady-state of water-based weathering and erosion operating during non-fire conditions. Palaeofire records in South Australia reveal evidence of episodic fire-related erosion events recorded in upland peat bogs of the Adelaide Hills[47]. This

supports the notion that sediment mobilisation in vegetated fire-prone regions is triggered by fire events that expose the soil directly to subsequent rainfall events leading to distinct pulses of hillslope erosion and sediment movement[51,52]. Statistical analysis of 223 fire records across Australia indicates climatic variations control fire regimes, with colder periods characterised by less burning and warmer intervals by more[48]. However, the records did not show any significant change to fire regimes with the arrival of humans and fire-stick farming techniques some 50–60 thousand years ago. This suggests that climate is the primary driver of fire regimes. We suggest that fire is not only important in terms of mobilising sediment reservoirs[52] through the temporary removal of binding vegetation cover, but also in generating new sediment, particularly around the peripheries of inselbergs and along escarpment fronts in hot, fire-prone continental interiors such as central Australia. Accurate long-term landscape models would need to adjust the fire recurrence interval according to climatic changes associated with glacial and interglacial cycles and possibly the more intense fire regimes associated with modern land management practices and a warming climate. Australia has just experienced an unprecedented fire season with over 18.6 million hectares burnt, mostly along eastern Australia[53]. The volume of rock fragments spalled from rock exposures after a mega-fire event of this severity represents a massive departure from "normal" background rates of bedrock erosion as well as subsequent soil/sediment mobilisation during rainfall events following the fires. To put this in perspective, if a conservative estimate of just 0.1% of the 18.6 million hectares of burnt areas across Australia in 2019–2020 were rocky outcrops affected by fire-spalling, on average 1 cm thick; then this would equate to over 5 million tonnes of spalled rock during the 2019–2020 fire season. If 1% of the burnt area spalled 2 cm of rock then the amount of sediment produced is closer to 100 million tonnes. Accurate estimates will vary depending upon the terrain, the outcrop, rock-type, fire intensity and whether the fires were in mountainous regions with abundant rocky outcrops.

We estimate that rates of lateral erosion due to fire-spalling in dry, fire-prone regions of central Australia may be up to 3.2 mm yr$^{-1}$ which is not a lot less than the rates of vertical fluvial or glacial erosion (1–10 mm yr$^{-1}$) measured in tectonically active mountains[13] and is four orders of magnitude (10,000 times) greater than the vertical rates of denudation (0.0003 mm yr$^{-1}$) operating at the top of inselbergs. This differential erosion pattern is responsible for maintaining steep-sided inselbergs until complete planation. Fire-spalling acting on a 2 m high rock face around the periphery of an inselberg, such as Uluru, would appear to generate about 64 times more sediment than the slow rates of vertical denudation acting across the top of the entire inselberg. Fire-spalling only works laterally at ground level where there is sufficient fuel load to generate fires of enough severity to spall fresh rock. Fire will not have any significant effect on vertical denudation in a flat environment because soil and regolith insulate and protect the underlying rock from the effects of surface fires. Fire-spalling denudation and sediment production cease once the inselbergs are flattened however subsequent fires do remove vegetation and aid in the mobilisation of surface sediment via water or wind activity[52].

The process of fire-spalling requires more quantitative studies including detailed photogrammetry and LIDAR surveys of flared slopes before and after fire events to establish the volume of mass wasting during fire events. Additional exposure age studies including cosmogenic exposure ages[14] and newly developed luminescence surface exposure dating[54,55] of exposed surfaces around the flanks of inselbergs might also help to establish the timing of past fire-spalling events and build more robust paleofire records. Likewise, lichenometry[56,57], which is used to date

Holocene glaciated terranes or landslides, might provide a method of calculating the age since the last fire-spalling event on a rock surface given that lichens and mosses are completely burnt and removed by fire-spalling at ground level but survive a few metres above the spalled rock surface (Fig. 4).

Recognition of fire-spalling as a major mechanism of weathering has relevance to the debate surrounding the formation of the steepened and flared margins of inselbergs. Two conflicting hypotheses for the origins of inselbergs have long been aired in multiple papers by King[58–60] and Twidale[21,61,62] and Twidale and Bourne[63,64] as well as in many other publications. King favoured inselberg formation by the parallel retreat of slopes in bedrock (pediplanation) over vast distances following initial valley incision. On the other hand, Twidale invoked a two-stage model of deep weathering of vulnerable rock and the stripping of the weathered material upon uplift, thereby exposing the weathering front and the more resistant unweathered bedrock as depicted in Fig. 6a. King noted the absence of deep weathering in some inselberg areas, as in the Valley of a Thousand Hills in Natal, South Africa, as well as incongruities between the depths of weathering and the heights of the inselbergs. Following this, Twidale introduced the notion of the episodic exposure of inselbergs through multiple stages of deep weathering and stripping, with the flared slopes representing the mere retouching of the outer flanks of the inselbergs. It was argued that hypotheses had been advanced and tested against the field evidence before arriving at the above theory, which was considered to be most likely to be correct. However, it was conceded that the conclusion could be modified if further evidence demonstrated its inadequacy[6].

We consider that the role of fire-related weathering in the development of inselbergs and flared slopes may be a missing link in the previous theories proposed (Fig. 6b). Fire is a very effective mechanism of weathering, even of fresh bedrock, while the rates of weathering suggested are commensurate with the large-scale parallel retreat of slopes, and the modification and further development of the margins of inselbergs. Fire sculpting is an important agent of geomorphic change in any fire-prone environment but is particularly evident in hot, dry, non-glaciated and tectonically inert continental regions like Australia, where we hypothesise fire-spalling dominates over fluvial and chemical weathering to create flared slopes and steep-sided inselbergs. The role of fire-spalling requires consideration in the lateral erosion of inselbergs, scarp retreat, sediment production and landscape evolution models in fire-prone environments. The presence of flared slopes around the periphery of inselbergs and escarpments in arid environments where fluvial and chemical weathering processes are only operating slowly is a potential indicator of some of the most intensely, fire-affected regions on Earth and the role of fire as an agent of long-term weathering and erosion on a flammable planet.

## Methods
This study is based on detailed field observations presented in the context of existing findings from published literature to develop a working hypothesis of fire-induced rock spalling. Field measurements of spalled flake widths and spalled surface areas were taken using an 8 m tape measure to the nearest mm and a portable luggage scale was used to measure the weight of spalled rock to the nearest 0.01 kg. All measurements are described in the text or annotated on the figures. Data sources are provided in captions.

## Data availability
Data sharing not applicable to this article as no datasets were generated or analysed during the current study. Source data for Fig. 1 is provided with the paper. Locations (latitude and longitude WGS84) of new, un-named field sites pictured in Figs. 4 and 5 are provided in figure captions.

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

## Acknowledgements

We would like to acknowledge the traditional owners of the lands covered in this paper and to whom these rock features have been central to their cultural traditions. We thank Colin Murray-Wallace and Wanchese Saktura for their discussions and Kai Buckman for his mathematical assistance. This research was supported by GeoQuest Research Centre and sabbatical support provided at the University of Wollongong that made fieldwork possible.

## Author contributions

S.B. conceptualised the idea, collected field observations, devised the fire-spalling formula/hypothesis and wrote the article. R.M. provided specific input into fire-related variables, fire history map production (Fig. 1a), background into post-fire soil erosion, collection of field data and revisions of the paper. R.B. provided background context to flared slope development and the broader historical context of landscape evolution models. He observed and provided pictures of inselbergs in Fig. 3a–f and revised the paper.

## Competing interests

The authors declare no competing interests.
