## [Peer Review File · Nature Communications]

REVIEWER COMMENTS

Reviewer #2 (Remarks to the Author):

Review of "Flared slope and inselberg formation via fire-induced rock spalling"

General comments:

Rock spalling from thermal exposure during wildfires is recognised by many as an important weathering agent, particularly in Australia. In this study, the authors introduce spalling as a primary agent of flared slope development at ground level at the periphery of inselbergs.

The study does provide a novel view regarding the formation of iconic landscape features which is of interest to specialists in the field of geomorphology and therefore meets the basic criteria for publication in Nature Communications.

The work is primarily based on theory, qualitative field observations, some published data and the development of a simple formula for spalling erosion. The work is therefore conceptual rather than quantitative and based on simple arguments rather than detailed analysis. Its advance lies in presenting a new hypothesis of how flared slopes might develop that is likely to trigger future research, rather than testing this hypothesis with new data. Whether or not this alone is sufficient to warrant publication in Nature Communications remains debatable. The study would certainly be stronger if data had been collected that underpins the novel view presented here.

The authors suggest that "fire-spalling is a significant driving mechanism of physical weathering in arid, fire prone environments" yet their examples of intensive fires and measurements of actual spalling are from the temperate East and South East of Australia (Figs. 3 & 4). Figure 1 d is labelled a spalling from fire, but the current line of vegetation (fuel) is quite some distance away. Is there specific evidence to suggest this is actually pyrogenic spalling?

The theory that in dry regions, the denser vegetation (trees and shrubs) surrounding inselbergs provides enough thermal energy during fires to cause spalling (Fig. 5) is conceptually compelling, but it would be useful if more evidence could be presented that this narrow zone of denser vegetation is commonly burning (and at sufficient intensity – i.e. rate of energy release) during landscape fires around e.g. Uluru or Wave Rock (e.g. fire scars in tree rings, charcoal records or other evidence). It is of course conceivable that the fire regime since human occupation in Australia is not a reflection of the fire regime and hence spalling prior to human modification of fire.

In any case, specific measurements that indicate the age of the flared slope (cause by spalling) in comparison to the areas above that have not be exposed to spalling would support the theory (e.g. Cosmogenic nuclides; Schmidhammer surface hardness measurements). That said, their support would not very specific exposure ages do not necessarily imply as specific weathering process. Perhaps thermal luminescence dating of sand grains or thermally induced changes in mineral magnetic properties in sediment below the flared slopes might be the best way forward to conclusively detect heating.

Whether or not more supporting evidence can be produced here or will be left for future studies, there is one aspect of the overall conceptual argument that I find needs more consideration. If fire-induced spalling is a key process in delivering debris on the flanks of inselbergs, this material still needs further breakdown and transport from the flanks in these flat landscapes. Otherwise there would be notable spalling debris mounds around the inselbergs (in addition to some rockfall not directly from spalling as seen in Fig. 1 e and f). If in situ chemical and physical weathering – albeit slow – is able to break this spalling debris down so that it can be easily removed (as fine particles or in solution), it seems logical that these non-pyrogenic processes may also able to lead to substantial weathering direct on the flanks of the inselbergs (without fire as a key process).

In short, few would doubt that fire is an important rock weathering process in fire-prone areas with high fuel loads (see photos in attached report), but the study does not go much beyond proposing a very interesting alternative hypothesis to the genesis of flared slopes in more arid regions, that I hope the authors and the wider geomorphological community will explore further.

Specific comments:

Abstract:

"Fire sculpting is an important agent of change in hot, dry, tectonically inert continental regions like Australia, which experience regular, intense and extended fire regimes as observed in the 2019-2020 Australian fires"

I am not sure how much the fires in the hot, dry interior of Australia are representative of the 2019/20 fires. This may be (unintentionally) misleading in that most of the reported impacts of the extreme 19/20 fires were in the temperate eucalypt forests of SE Australia (see also images below). This temperate forest fire regime differs in many respects from that of the hot and dry regions of Australia.

363: "Fire spalling has yet to be fully recognised as a mechanism of physical weathering and erosion in most erosion and landscape evolution models". It has certainly been highlighted in various studies incl. the review of ref. 47, but it may indeed still not be considered sufficiently in models.

558: should be "Zeitschrift für Geomorphologie"

Examples of cave (tafoni) development used in Fig. 1 f & g. Tafoni development, whilst still somewhat enigmatic, has been examined in many environments and explained by non-pyrogenic processes. Their use here detracts from, rather than supports the case made here. Their floors are generally vegetation free and pyrogenic spalling is therefore not a viable mechanism for their continuous inward development.

I trust you find the comments above useful and constructive.
Prof. Stefan Doerr

Reviewer #3 (Remarks to the Author):

Flared slopes are a particular feature of some inselbergs in arid lands, including Uluru and Katter Kich in Australia. The development of flared slopes has been attributed either to subsurface weathering followed by erosion or to subaerial forces of slope retreat. Widespread fires in Australia in 2019-2020 led to extremely high temperatures being applied to exposed rock surfaces. This resulted in spalling of mainly near-ground surfaces on boulders and rock faces. Following field observations and measurements of spalls on sampled surfaces, it is proposed that such fire-induced spalling is the primary mechanism for development of flared slopes. The process would thus have implications for models of landscape evolution in fire-prone environments.

The claims made in this paper would be of interest to those proposing different origins (subsurface or aerial) for flared slopes and may generate continuation of this discussion.

Comments and questions -

There would be little debate about the capacity of fire to generate spalling in exposed rocks which are cohesive and quartz-bearing, or that the immediate effects would be most apparent near ground level where radiant heat from fire would be concentrated. The following comments and questions would

benefit from the author/s consideration.

- More detailed discussion could be provided on the reasons for flared slopes being present on only some margins of individual inselbergs even when vegetation (fire fuel) surrounds the entire outcrop.
- It would be useful to make clear whether fire-induced rock spalling is viewed as a weathering or an erosional process (or if such nomenclature is relevant in the present context).
- The base of the overhang in Figures 2 (b) and 2 (f) is separated from vegetation by several metres of flat to gently sloping bare rock. Does this absence of a direct vegetation/rock interface influence the effectiveness of the proposed fire-spalling mechanism of slope retreat over long time frames?
- The height of vegetation and of flared slopes at Uluru currently coincide (Fig. 5). To what extent is this a fortuitous outcome of the (geologic) observation time – i.e. has vegetation remained the same over any proposed duration of development of the flared margins of the inselberg? The impact, if any, of changes to vegetation over lengthy time frames might be considered.
- In discussing rates of erosion and the evolution of inselberg reduction in an arid environment, a figure illustrating the role of fire in the sequence of landscape planation would be desirable. This would enhance the descriptions given.
- Is there evidence of past fire spalling on flared slopes? If the event scenario described in equation 1 occurs (a fire interval of 50 years and 20% of the near-ground rock surface affected), then previous fire spalling should be identifiable.
- In order to provide an international context, the inferred rates of reduction of upper surfaces of inselbergs reported in the literature could be listed in a table along with estimates presented for the Australian case. This would strengthen the points being made about rates of breakdown by fire spalling. Mention could also be made of whether flared slopes occur in arid environments other than those reported here – is the feature being described unique to Australia? Or have similar relationships between fire and flared slopes been identified on inselbergs in other continents?
- The example of Uluru (arkose sandstone) was given for sediment production rates using equation 2. In the example the rock density value for granite was used rather than that for sandstone. As field data were collected for granites it is logical to use these values, but the rock-type substitution in the equation needs to be mentioned/ justified.
- Minor: The typo describing a flared slope as a cave (paragraph 1) needs correcting – presumably the reference to cave was meant to follow 'tafoni'.
- Minor: the term 'morros' is used for 'overhanging flared slopes' – is the term used for inselbergs, or only their flared slopes? A specific reference would be useful here given the journal's international readership.
- The paper may be more suited to a geomorphology journal, with emphasis either on the feature of rock spalling or on the landscape evolution component (the major outcome of the fire-induced rock spalling described).

Detailed Response to Reviewers and Editors Comments

Please find the following responses (blue italics) to the reviewers' comments (black).

Reviewer #2 comments:

Review of "Flared slope and inselberg formation via fire-induced rock spalling"

General comments:

Rock spalling from thermal exposure during wildfires is recognised by many as an important weathering agent, particularly in Australia. In this study, the authors introduce spalling as a primary agent of flared slope development at ground level at the periphery of inselbergs. The study does provide a novel view regarding the formation of iconic landscape features which is of interest to specialists in the field of geomorphology and therefore meets the basic criteria for publication in Nature Communications.

Thank you for acknowledging this as a new and novel hypothesis. This idea is a massive departure from the current thinking which revolves around subsurface weathering and fluvial excavation (Twidale) which is why we submitted to Nature Comms which encourages new and innovative ideas. A few (not many!) have recognised that fire breaks down rock surfaces, but our point is that fire has only been considered a minor, occasional phenomena that only sporadically affects rocks without any real impact on broader landscape evolution or long-term weathering. Geomorphologists have tended to look at fire-related rock spalling in a modern, short-term context, e.g., the effect on cave art, without projecting the longer-term effects of fire on landscape evolution over 100's of thousands or millions of years. The main thrust of our hypothesis is that inselbergs and flared slopes and rounded granite tors of inland Australia probably didn't evolve via subsurface weathering but via fire spalling. Fire spalling is not recognised by many as an important weathering agent – more as an occasional, rare and isolated phenomena of little broader importance.

The work is primarily based on theory, qualitative field observations, some published data and the development of a simple formula for spalling erosion. The work is therefore conceptual rather than quantitative and based on simple arguments rather than detailed analysis. Its advance lies in presenting a new hypothesis of how flared slopes might develop that is likely to trigger future research, rather than testing this hypothesis with new data. Whether or not this alone is sufficient to warrant publication in Nature Communications remains debatable. The study would certainly be stronger if data had been collected that underpins the novel view presented here.

We agree that this new hypothesis will spawn new research which is why we submitted the manuscript to Nature Comms which aims to represent important advances of significance to specialists within each field. This is a new and novel idea that will trigger new research particularly given the horrendous summer of fires experienced in Australia in 2019-20. We acknowledge that there is much more quantitative data that could be collected but these are specific studies for the future that will take years to complete. Our photographic data of case

studies collected during field trips post-fire examples across a number of different geological and environmental regimes along with the existing studies of others is enough to conclude that fire spalling is obviously a ubiquitous process operating on rock surfaces during fires and needs to be considered in broader landscape models.

We are planning future detailed photogrammetry and controlled heating experiments of various rock types to better understand the nature of fire spalling. Certainly, the proponents of sub-surface weathering models for flared slope development will want to counter our hypothesis and thus new data from different perspectives will be presented and further our scientific understanding of inselberg and flared slope formation.

To clarify, our model is primarily based on our own field observations but reinforced by observations from other studies in order to develop a working hypothesis so we are following the correct scientific process rather than applying an existing “theory” to a particular setting – it is certainly not a theory as such – I wouldn’t be so arrogant as to think the hypothesis is completely tested and foolproof and therefore worthy of “theory” status). The idea sprang to mind when walking around Uluru and observing the flared slopes which I’d read about previously and discussed with co-author Bob Bourman. A recent fire had charred the nearby trees around Uluru and the height of the charred trunks was exactly the same height as the adjacent flared slope. That’s when the idea hit! The flared slope also had a blackened surface apart from the small spalled patches that had popped off during or just after the fire. The environment around Uluru is now heavily modified with a cleared walking track between the rock and vegetation which would reduce the effects of fire spalling. The fire itself may have been a backburn in response to an approaching fire front in 2012 and therefore relatively low intensity but despite the modified environment and low intensity of the fire there was still evidence of fire spalling along the flared slope.

Our study is based on clear observations of fire induced rock spalling from many sites now in which we have provided mostly qualitative (photographic) evidence but also some quantitative evidence (spalled surface areas and weight of spalled material). Many new ideas start with simple observations but are further tested by sophisticated analyses in order to verify what is sometimes quite obvious. The best ideas appear to be quite obvious when pointed out clearly and yet they often go unnoticed for a long time because they sometimes require a bit of lateral thought to link together two seemingly disparate processes. Fire is not commonly considered in terms of geological processes and yet it is a process that is unique to Earth with its abundance of flammable material at the surface due to the presence of life.

The simplest arguments are often the best (Occam’s Razor) and a picture tells a thousand words! We certainly plan on making more detailed, sophisticated measurements of fire-induced rock spalling at specific locations across Australia and other countries in order to better quantify the amount of rock spalled according to the intensity of the fire etc but these measurements will just provide more accurate constraints on the amount and rate of spalling in different environments and rock types. The fact that spalling is happening during fires is obvious from our field observations and our equation provides a foundation for measuring the key variables that determine the rate of fire spalling erosion in any setting.

Yes. The initial idea was developed after observations of the fire and flared slopes at Uluru and we initially focussed on arid landscapes. However, when the 2019-20 fires along eastern Australia occurred we confirmed that fire spalling was common within intensely burnt areas and in differing geologies eg granite vs sandstone. We included examples from eastern Australia because it was such a compelling example of fire induced spalling and really emphasised that pyrogenic spalling occurs in any fire-prone environment, not just arid regions of Australia.

The authors suggest that “fire-spalling is a significant driving mechanism of physical weathering in arid, fire prone environments” yet their examples of intensive fires and measurements of actual spalling are from the temperate East and South East of Australia (Figs. 3 & 4). Figure 1 d is labelled a spalling from fire, but the current line of vegetation (fuel) is quite some distance away. Is there specific evidence to suggest this is actually pyrogenic spalling?

Flares are not restricted to deserts and in past climates some temperate areas were more desert like. We simply make the point that fire spalling will occur in any fire-prone environment.

Yes, the flared slope surface is blackened from the fire but the surface beneath the spalled flakes is soot-free and therefore must have popped off during the later stages or just after the fire passed through in Dec 2012 – we observed the spalling in May 2013. The examples we show from the granites and sandstones clearly demonstrate the erosive power of fire spalling but we hope to visit some of the classic inselbergs in central Australia before and after fire events in order to document the spalling process in more detail.

The theory that in dry regions, the denser vegetation (trees and shrubs) surrounding inselbergs provides enough thermal energy during fires to cause spalling (Fig. 5) is conceptually compelling, but it would be useful if more evidence could be presented that this narrow zone of denser vegetation is commonly burning (and at sufficient intensity – i.e. rate of energy release) during landscape fires around e.g. Uluru or Wave Rock (e.g. fire scars in tree rings, charcoal records or other evidence). It is of course conceivable that the fire regime since human occupation in Australia is not a reflection of the fire regime and hence spalling prior to human modification of fire.

Some good ideas but Australian vegetation is so adapted to fire and drought that it doesn't have annual growth rings and fire affected tree trunks tend to shed their burnt bark in a similar fashion to rock spalling and thus they don't always preserve a fire scar. European deciduous trees are very different. There might be potential to identify fire scars in tree rings of larger trees around the edges of inselbergs and potentially date the charcoal extracted from each fire scar, but we wouldn't be able to rely on the annual growth ring counting for Australian vegetation. There are very few charcoal records obtained within Australia. We have provided some for a peat bog in Adelaide and there are a few others, but very few for inland Australia. It is a topic that needs much more study and we would certainly like to look at charcoal records in some of the salt lakes for inland Australia.

We covered the point of human occupation which was specifically addressed by Mooney et al 2011 who showed that the arrival of humans some 65 ka did not have a statistical impact on fire records. Modern environmental changes over the past 200 years associated with clearing and grazing have probably changed fire regimes substantially but again this is not well studied in the sedimentary record.

In any case, specific measurements that indicate the age of the flared slope (cause by spalling) in comparison to the areas above that have not be exposed to spalling would support the theory (e.g. Cosmogenic nuclides; Schmidt Hammer surface hardness measurements). That said, their support would not very specific exposure ages do not necessarily imply as specific weathering process. Perhaps thermal luminescence dating of sand grains or thermally induced changes in mineral magnetic properties in sediment below the flared slopes might be the best way forward to conclusively detect heating.

Yes, the cosmogenic studies available suggest old at the top of inselbergs and extremely variable around the flanks as stated in the manuscript. We cover this aspect early on in the manuscript. Whilst it would be good to undertake more cosmogenic studies, specifically with fire-spalling in mind, it would probably only replicate the existing studies and take many years of analysis.

Spalled rock becomes very crumbly after heating and peels off the rockface in sheets. You can easily crumble spalled sheets of what was fresh granite in your hand following the fire spalling event. A Schmidt Hammer test might be useful in terms of differentiating fresh un-spalled surfaces from surfaces that have been burnt and weakened but remain attached to the rock surface.

Luminescence dating techniques hold some potential for determining the last time a rock surface was heated to high temperatures to reset the luminescence in minerals such as quartz and feldspar. However, we would have to establish how deeply into the rock the heat is conducted and this would require extensive experimental work. Researchers at our institution are currently working on this aspect so we may be able to develop this in the future and potentially establish very young TL ages around the flanks of the inselberg and progressively older ages towards the summit of the inselberg in a manner similar to cosmogenic studies.

Mineral transformations might be a useful way of determining if a surface has been heated substantially or not. We are aware of the effect of fire in terms of transforming iron oxides to maghemite. There may be other minerals such as apatite or micas that anneal or become isotopically reset by the high fire temperatures but whether the short duration of a fire is enough to fully reset these minerals is a study that would require detailed experimentation.

Whether or not more supporting evidence can be produced here or will be left for future studies, there is one aspect of the overall conceptual argument that I find needs more consideration. If fire-induced spalling is a key process in delivering debris on the flanks of inselbergs, this material still needs further breakdown and transport from the flanks in these flat landscapes. Otherwise there would be notable spalling debris mounds around the inselbergs (in addition to some rockfall not directly from spalling as seen in Fig. 1 e and f). If

in situ chemical and physical weathering – albeit slow – is able to break this spalling debris down so that it can be easily removed (as fine particles or in solution), it seems logical that these non-pyrogenic processes may also be able to lead to substantial weathering direct on the flanks of the inselbergs (without fire as a key process).

No. The spalled rock has been broken down by the fire and is very friable after spalling. The spalled sandstones in the Blue mountains tend to crumble into sand when you pick them up. Granite remains more robust but is also considerably weakened after spalling off the surface. The spalled sheets are easily broken into individual grains after fire spalling or by subsequent fires and then easily transported by wind or water which can flow very rapidly off the impermeable inselbergs during occasional storms and flash flooding. Uluru forms a distinct deflation depression around its margins due to the planar wind flow across the plains turning turbulent when it encounters the steep ~400 m rock jutting out from the plain around its periphery. The thicker vegetation around Uluru gives way to spinifex covered sand dunes several hundred metres from the rock.

It might be worth looking at the debris around inselbergs to quantify the mineralogy which would reflect the degree of chemical weathering – i.e. is it quartz-rich which would reflect extended periods of chemical weathering or is there an abundance of less weatherable feldspars and mafic minerals derived from the fresh granite or arkose sandstone (Uluru) that suggest that the sediment has not travelled far from the source (inselberg) and not undergone significant chemical weathering since it was dislodged from the rock source? This is actually a useful test for future work but again would take considerable time to complete.

In short, few would doubt that fire is an important rock weathering process in fire-prone areas with high fuel loads (see photos in attached report), but the study does not go much beyond proposing a very interesting alternative hypothesis to the genesis of flared slopes in more arid regions, that I hope the authors and the wider geomorphological community will explore further.

It is really great to hear that Reviewer 2 agrees that there is little doubt of the importance of fire in rock weathering in fire prone areas. We respectfully disagree with the statement that this study does not go beyond proposing a model for flared slope genesis. Flared slopes are an important geomorphic indicator of repeated fire spalling on a rock surface and this is where our research started but we have provided numerous examples of fire spalling following the 2019-20 eastern Australia fires that show that fire spalling is ubiquitous across fire prone environments and occurs at rates that are geologically rapid. We incorporate those rates into an equation that can be applied to landscape models and has predictive qualities in terms of measuring escarpment retreat and sediment production over long time scales. We emphasise that no one has really considered fire spalling as anything other than a local phenomenon and certainly no-one has linked fire with flared slope and inselberg formation previously. We certainly look forward to the broader geomorphological community testing and refining our fire-spalling model more rigorously in the future.

Specific comments:

Abstract:

“Fire sculpting is an important agent of change in hot, dry, tectonically inert continental regions like Australia, which experience regular, intense and extended fire regimes as observed in the 2019-2020 Australian fires”

I am not sure how much the fires in the hot, dry interior of Australia are representative of the 2019/20 fires. This may be (unintentionally) misleading in that most of the reported impacts of the extreme 19/20 fires were in the temperate eucalypt forests of SE Australia (see also images below). This temperate forest fire regime differs in many respects from that of the hot and dry regions of Australia.

We take the point here. Our concept of fire spalling was initially developed after viewing the flared slopes around Uluru and we concentrated on inselbergs in arid Australia, but the concept grew to include any areas affected by intense fire as observed during the eastern Australia 2019-20 fires. Our point is that any environment affected by fires will be prone to fire-induced rock spalling as our pictures clearly show! The difference is that fluvial and chemical weathering is much more subdued in arid regions and thus the effects of fire spalling in the form of flared slopes and steep sided inselbergs is much more prominent in arid Central Australia than temperate coastal areas of eastern Australia.

Yes! Temperate forest fire regimes are different from dry Central Australia but both environments burn as seen by the numerous fire scars in aerial views around Uluru and as observed along eastern Australia in 2019-20. Wherever we have intense burning we have potential for fire-induced rock spalling. Different fire regimes will give different rates of fire induced erosion, but our equation takes that into consideration.

363: “Fire spalling has yet to be fully recognised as a mechanism of physical weathering and erosion in most erosion and landscape evolution models”. It has certainly been highlighted in various studies incl. the review of ref. 47, but it may indeed still not be considered sufficiently in models.

Yes, we agree 100% - indeed this is the very point of our paper. Fire induced rock spalling has only been considered a local phenomenon in the past and not considered in broader landscape models.

558: should be “Zeitschrift für Geomorphologie”

Thank you. Corrected in references.

Examples of cave (tafoni) development used in Fig. 1 f & g. Tafoni development, whilst still somewhat enigmatic, has been examined in many environments and explained by non-pyrogenic processes. Their use here detracts from, rather than supports the case made here. Their floors are generally vegetation free and pyrogenic spalling is therefore not a viable mechanism for their continuous inward development.

OK but tafoni are not really the focus of this paper – just pointing out that they are present and not really related to sub-surface weathering. We have deleted reference to tafoni to reduce unnecessary confusion.

Well we don't know how heat from the burning vegetation just a few metres outside the cave radiates into the cave and affects the cave surface. The caves are probably vegetation free because they have been occupied for the past 40-60 ka by indigenous people who also managed fuel loads around the inselbergs which were a natural site to occupy because of the presence of permanent water and abundant food and shelter. We also don't really know how campfires lit inside these cave shelters might have created ongoing spalling immediately above the contained fire. We have removed the mention of tafoni as it isn't critical to this manuscript.

I trust you find the comments above useful and constructive.
Prof. Stefan Doerr

Reviewer #3 (Remarks to the Author):

Flared slopes are a particular feature of some inselbergs in arid lands, including Uluru and Katter Kich in Australia. The development of flared slopes has been attributed either to subsurface weathering followed by erosion or to subaerial forces of slope retreat. Widespread fires in Australia in 2019-2020 led to extremely high temperatures being applied to exposed rock surfaces. This resulted in spalling of mainly near-ground surfaces on boulders and rock faces. Following field observations and measurements of spalls on sampled surfaces, it is proposed that such fire-induced spalling is the primary mechanism for development of flared slopes. The process would thus have implications for models of landscape evolution in fire-prone environments.

The claims made in this paper would be of interest to those proposing different origins (subsurface or aerial) for flared slopes and may generate continuation of this discussion.

Comments and questions -

There would be little debate about the capacity of fire to generate spalling in exposed rocks which are cohesive and quartz-bearing, or that the immediate effects would be most apparent near ground level where radiant heat from fire would be concentrated. The following comments and questions would benefit from the author/s consideration.

- More detailed discussion could be provided on the reasons for flared slopes being present on only some margins of individual inselbergs even when vegetation (fire fuel) surrounds the entire outcrop.

The presence of flared slopes on the margins of some inselbergs and not others will be a function of rock type, fire regime, climate, vegetation type, terrain (high relief or flat), rates of chemical weathering and landscape stability over the past several thousand years. Generally, the thickest vegetation occurs on the shady (southern) side where water runoff likely to soak in and provide water for plant growth for longest time. Also, the prevailing direction of approach of fire could influence the position of some flares. Certainly, landuse has changed dramatically in the past few hundred years since European settlement and thus many of the inselbergs either don't have the vegetation that once surrounded them due to

land clearing and farming or in some cases the opposite has occurred and the conservation of the area as national park has allowed vegetation to regrow unabated and thus some segments of the inselberg margins are shrouded in thick vegetation. We suspect that indigenous cultures probably had permanent settlements around many of these rock features due to the presence of permanent water and abundant food and shelter and thus they probably managed the vegetation through fire-stick farming practices. Thus, it is difficult to discuss or infer the relationship between flared slope and existing vegetation patterns that may have been very different a few hundred years ago.

- It would be useful to make clear whether fire-induced rock spalling is viewed as a weathering or an erosional process (or if such nomenclature is relevant in the present context).

Weathering is the breakup of rock. Erosion its transport – mass movement, water, wind, ice. We suspect that fire induced rock spalling is both. It is a form of physical weathering because it breaks up the original rock into flakes of friable rock and it is erosional because the flakes drop off the rock face and can easily roll down slope or be further transported (eroded) by fluvial or aeolian activity.

- The base of the overhang in Figures 2 (b) and 2 (f) is separated from vegetation by several metres of flat to gently sloping bare rock. Does this absence of a direct vegetation/rock interface influence the effectiveness of the proposed fire-spalling mechanism of slope retreat over long time frames?

Currently in its present form as described it would inhibit vegetation growth and therefore reduce the effect of fire-spalling but we don't know that there has always been a gap between the vegetation and rock. It is likely that the vegetation has, in part, been cleared from the rock recently because these are such tourist attractions now. The radiant heat from an intense fire is enough to travel a few metres and still spall the rock as seen in the pictures of Uluru where spalling occurred even though a 3-4 m wide path buffered the rock from the nearest vegetation.

- The height of vegetation and of flared slopes at Uluru currently coincide (Fig. 5). To what extent is this a fortuitous outcome of the (geologic) observation time – i.e. has vegetation remained the same over any proposed duration of development of the flared margins of the inselberg? The impact, if any, of changes to vegetation over lengthy time frames might be considered.

Vegetation has undoubtedly changed over time and since humans arrived and started occupying the caves and then clearing the vegetation. However, the prevailing vegetation in these arid areas is small trees and shrubs and spinifex on the flat plains. It is only next to shaded, sheltered areas that some larger trees tend to grow due to the presence of shallow groundwater sources. You can see this in less developed, less visited inselberg sites such as Walga Rock in W.A. but also around well-known features such as Wave Rock. In each of these examples, the vegetation thickens around the inselberg, reaching in towards the flared slopes and coincides with the height of the concavity. The presence of vegetation is controlled by the presence or absence of soil for the vegetation to take root in – bare rock

doesn't allow this. We agree that it would be useful to make more detailed studies of vegetation changes over time at specific localities but again this is for future detailed case studies.

- In discussing rates of erosion and the evolution of inselberg reduction in an arid environment, a figure illustrating the role of fire in the sequence of landscape planation would be desirable. This would enhance the descriptions given.

I have added a topographical cross-section through Kata Tjuta- Uluru-Mt Connor that shows the previous paleosurface some 400m above the current ground surface. I think this gives some perspective to long-term landscape evolution that we discuss. I'm not sure that a sequential landscape planation model is necessary as the actual topographic profile illustrates the faster lateral rates of erosion compared to the vertical rates but I'm happy to provide one if you think it is worthwhile.

- Is there evidence of past fire spalling on flared slopes? If the event scenario described in equation 1 occurs (a fire interval of 50 years and 20% of the near-ground rock surface affected), then previous fire spalling should be identifiable.

Not really because it is removed as spalled sheets. The only evidence is the presence of black soot on the rock surfaces that haven't spalled off and the broken up spalled sheets on the ground. However, the soot washes off over time and is covered over by algae or lichens. Lichenometry is something we are considering in order to establish the age of particular surfaces and potentially establish a short-term fire record for a particular rock face but lichenometry is in its infancy in Australia and would require much more research and testing.

- In order to provide an international context, the inferred rates of reduction of upper surfaces of inselbergs reported in the literature could be listed in a table along with estimates presented for the Australian case. This would strengthen the points being made about rates of breakdown by fire spalling. Mention could also be made of whether flared slopes occur in arid environments other than those reported here – is the feature being described unique to Australia? Or have similar relationships between fire and flared slopes been identified on inselbergs in other continents?

We have used and stated the inferred rates of erosion of upper surface of inselberg from the cosmogenic studies by Bierman. I'm not sure that tabulating these existing results adds much to this paper. The key observation by Bierman is that the flanks of inselbergs show variable but faster rates of erosion which is in line with our hypothesis of lateral attack by fire spalling at ground level. There are flared slopes elsewhere around the world but not as well developed. This is probably because Australia is a particularly fire-prone continent.

- The example of Uluru (arkose sandstone) was given for sediment production rates using equation 2. In the example the rock density value for granite was used rather than that for sandstone. As field data were collected for granites it is logical to use these values, but the rock-type substitution in the equation needs to be mentioned/ justified.

The compacted and slightly metamorphosed arkose sandstone that makes up Uluru is about the same density as a granite, so we have used the same value and stated this in the text. It isn't going to affect the calculations in any significant manner but we have noted the density value assumptions in the text.

- Minor: The typo describing a flared slope as a cave (paragraph 1) needs correcting – presumably the reference to cave was meant to follow 'tafoni'.

We have removed the terms 'cave' and 'tafoni' to reduce unnecessary confusion. We just use the term 'flared slope'.

- Minor: the term 'morros' is used for 'overhanging flared slopes' – is the term used for inselbergs, or only their flared slopes? A specific reference would be useful here given the journal's international readership.

We have removed the terms tafoni and morros as they are not important to the manuscript and we avoid any technical confusion.

- The paper may be more suited to a geomorphology journal, with emphasis either on the feature of rock spalling or on the landscape evolution component (the major outcome of the fire-induced rock spalling described).

We respectfully disagree with this evaluation and think that subsequent, quantitative, case studies are more suited to journals such as Geomorphology. Fire induced rock spalling is a newly identified process with novel applications in landscape evolution models. We illustrate this point by applying it to the long-standing but, in our view, incorrect model of flared slope development. We suspect this hypothesis will initiate many new studies particularly given the focus on wildfire in a changing climate.

Finally, we have included some additional discussion on the alternative models as to inselberg formation presented by King and later by Twidale et al. These are fundamental questions to geomorphologists and we believe that the introduction of fire as a weathering process helps to answer some of the long-standing questions regarding landscape evolution and inselberg formation.

REVIEWER COMMENTS

Reviewer #2 (Remarks to the Author):

The authors have provided a very detailed response to both reviewers' comments, although it would have helped to have identified the line numbers where specific modifications were made in the manuscript to address these points.

The manuscript is now more focused and the main arguments made are now better supported. What remains opaque in places is the distinction between the main focus of the manuscript (presenting the novel hypothesis that fire-induced rock spalling is a mechanism of physical weathering responsible for inselberg and flared slope development) and the overall importance of rock spalling as a weathering agent in fire prone environments. Whilst the authors correctly argue that fire-induced rock spalling "has yet to be incorporated into broad-scale landscape models", I find the claim (L511...) "it has yet to be recognised as an important and widespread mechanism of physical weathering and sediment production" not supported by evidence. In the rebuttal it is argued that "Fire spalling is not recognised by many as an important weathering agent – more as an occasional, rare and isolated phenomena of little broader importance". This could be seen as a matter of opinion, but the only reference supporting this claim is Nr24 (line 127), an article published in an international magazine (albeit peer-reviewed), whereas many studies also cited here have previously pointed out that fire is an important rock weathering agent, including one widely cited review article (Nr48), which highlighted even in the abstract that "Wildfire-induced weathering rates have been reported to be high relative to other weathering processes in fire-prone terrain". In my opinion this somewhat undermines the credibility of the work presented here. The authors' statement (L128-130) "Our observations of rock surfaces following wildfires are that fire related rock spalling is a commonly observed phenomena wherever high intensity fire has swept across rocky outcrops" supports what other studies have pointed out before.

I do suggest to make a clearer distinction in the abstract and main text between the novel idea that (i) fire-induced rock spalling is a mechanism of physical weathering responsible for inselberg and flared slope development and (ii) the wider relevance of fire-induced rock spalling in fire prone landscapes as a whole.

Specific comments:

Abstract: The statement "Fire sculpting is an important agent of geomorphic change in any fire-prone environment...but is particularly evident in hot, dry, non-glaciated and tectonically inert continental regions like Australia, where fire spalling dominates over fluvial and chemical weathering to create flared slopes and steep sided inselbergs". is based on the authors' hypothesis. This may well be the case, but this hypothesis has yet to be tested with empirical data and should therefore be rephrased to "Fire sculpting is an important agent of geomorphic change in any fire-prone environment...but is particularly evident in hot, dry, non-glaciated and tectonically inert continental regions like Australia, where we suggest/we hypothesise fire spalling dominates over fluvial and chemical weathering to create flared slopes and steep sided inselbergs".

L100, Fig. 2g and elsewhere: caves (incorrectly referred to Fig. 5 in the revised manuscript). Tafoni have been removed from the manuscript given that there was no convincing evidence for a pyrogenic origin (as acknowledged in the rebuttal). The same applies to caves which share many morphological features with tafoni. Retaining them in the manuscript adds nothing tangible to the arguments about flared slopes and instead confuses the issue. Shallow caves are found in many rock types and are often morphologically similar whether they occur in fire prone and virtually fire free environments.

L141: "Average flame-front residence time for eucalypt forest fuels is about 37 seconds but radiant heat and hot winds fanning out in front of the fire have the ability to pre-heat front the rock

surface and vegetation before and after arrival of the fire". This precise statement is based on one case study using experimental fires in jarrah forest of south-west Western Australia and is by no means representative of flame residence times for wildfires in eucalyptus forests as a whole – as implied here. For example, the fire behavior model developed based on these fires underestimated fire propagation in the 2009 Black Saturday fires.

L145 onwards: "Basalt is a high temperature volcanic rock with no quartz content. Fire spalling was minimal across most of the outcrops and generally consisted of dislodged phenocrysts. However, a few basalt outcrops adjacent nearby fallen burnt logs were intensely scorched and displayed thin (1-4 mm) spalled flakes of basalt indicating that fire spalling is not restricted entirely to quartz-rich lithologies." This is an important observation, which does point to a perhaps even more relevant point here and related to L141. Flame front residence times are generally short – in the order of seconds to a few minutes, EXCEPT where large woody fuels are present (termed 'down wood'). In mature eucalypt forests, dead down wood can burn or smoulder for hours and even days, providing ideal conditions for spalling (note the remnants of these in some of your images). However, this mechanism is likely to be much more prevalent in woodlands rather than arid or semiarid regions.

L295: I suggest "flame height" instead of "height of the fire" here to be more precise.

L486-492: It is clearly useful to put the effect of spalling into the context of erosion rates as a whole. In the 2019-20 fire season, the area burned in the SE Australian forest biome was ca 10 x that for a 'typical' fire season (<https://www.nature.com/articles/s41558-020-0716-1>) [although the area burned in Australia as a whole was relatively low]. Hence in this biome spalling would have affected a much larger area than is normal, although this may not be very relevant when considering the long term effect of fire enhanced erosion cycles as a whole.

Rather than providing a figure for the estimated mass of spalling over the area burned in this season (100 Mill tons), it would be much more meaningful to give a figure that enables comparison with other studies that examined fire induced erosion rates in this region of SE Australia (e.g. DOI: 10.1002/esp.1460), such as t/ha or mm/y (or mm/ky). I also doubt that the surface area of rock susceptible to spalling is equivalent to 1% of the entire land area burned. Spalling was clearly noticeable on some cliff faces for example in the Blue Mountains after the 2019/20 fires (as also depicted in Fig. 5), but this is one of the most cliff/outcrops rich areas in SE Australia. The Grose valley (Fig. 5g) has ca 20 km of cliffline in an area of 20 km². If as much as 1 m² per every m of cliffline had lost its surface from spalling in this fire, this would equate to 0.1% of the land surface affected by spalling (not 1% as used in the example by the authors), and this is likely to be one of the most cliff-dense and severely burned area within the 2019/20 area burned. In addition, 2 cm or thicker sheets of spalled rock can indeed be observed in some places burned in 2019/20 in the Blue Mountains, but this is likely to be at the high end of the spalling depth range. Thus, I suggest that the actual loss of material from rock faces in this event is likely to be orders of magnitude below that estimated (as an illustrative example) by the authors. A recalculation would bring this more in line with denudation and spalling estimates made in previous studies for this region (as summarised in DOI: 10.1002/esp.1460 and estimated by Humphreys et al. 2003. Some effects of fire on the regolith. In: Roach, I.C. (Ed.), *Advances in Regolith*. pp. 216–220.)

Stefan Doerr

Reviewer #3 (Remarks to the Author):

The authors have responded to the reviewers' comments and provided additional field observations and discussion on inselberg formation. Some of the issues relating to vegetation remain somewhat unconvincing but in the absence of conclusive evidence these arguments remain subject to individual interpretation (and do not negatively impact on publication). The comments below are for authors'

consideration only and do not require a formal rebuttal/reply.

The paper's title indicates that the authors consider fire-induced rock spalling is responsible for development of steep slopes on the margins of inselbergs. Are other mechanisms also involved, with the proviso that the flared slopes which appear on some inselberg margins may be distinctively and directly related to fire? Perhaps a word change in the title may be appropriate – Fire-induced rock spalling as a mechanism of physical weathering responsible for flared slope development on inselbergs (??)

It is implied that all rock outcrops are composed of relatively fresh rock. Equation 1 may benefit from a clear statement of this assumption, especially as the application of the model includes both wet temperate and hot desert environments – the factor of rock strength as well as rock type is relevant to W . In wetter climates, sandstone surfaces which are potentially more weathered, of lower strength and with a thicker weathering 'rind' may respond by disintegrating in a high intensity fire rather than by spalling. Regardless of the process, fire may lead to similar morphological outcomes on the outcrop but potentially different rates of surface change over time. Consider ' $W = \text{Average} \dots \text{Dependent on rock type (quartz content), rock strength, fire} \dots$ '

Very minor editing – Add reference 34 in paragraph commencing 'Although fire was first identified....back in 1927 34 it has yet...'

REVIEWER COMMENTS

Reviewer #2 (Remarks to the Author):

The authors have provided a very detailed response to both reviewers' comments, although it would have helped to have identified the line numbers where specific modifications were made in the manuscript to address these points.

A big thankyou to Stefan Doerr for his constructive comments. We have made a considerable effort to address all of the points raised and collected a substantial amount of new data to support our hypothesis. The revised manuscript includes line numbers and has been formatted in terms of headings and word limit requirements to fit the requirements of Nature Communications.

The manuscript is now more focused and the main arguments made are now better supported. What remains opaque in places is the distinction between the main focus of the manuscript (presenting the novel hypothesis that fire-induced rock spalling is a mechanism of physical weathering responsible for inselberg and flared slope development) and the overall importance of rock spalling as a weathering agent in fire prone environments.

Thankyou for helping to focus our arguments and we are glad our We have made a clearer distinction between the two aspects, particularly in the abstract, although the two aspects of flared slope development and fire-induced rock spalling as an agent of weathering are inextricably linked – this is at the very core of the hypothesis. We separated out our observations of fire spalling following the Black Summer fires to show that fire spalling is a common, widespread phenomenon during intense burning near rocky outcrops, but also point out that the initial idea was spawned by observations of fire spalling on flared slopes of Uluru following the 2012 fires there. We understand that eastern Australia is a different biome to Central Australia where we initially developed the idea of flared slope development but we do provide evidence of fire spalling that occurred on a flared slope at Uluru in 2012 and the abundant fire spalling observed in eastern Australia following the Black Summer fires just shows that fire-spalling occurs in any fire prone environment in which there is abundant rocky outcrops in close proximity to burning vegetation. Unfortunately wildfires occur episodically and randomly and so there is an opportunistic aspect to making relevant observations. As with Uluru in 2012, other prominent inselbergs and flared slopes will be affected by fire in the near future and we will know what to look for and how to measure the spalled surfaces. The Black Summer fires were unprecedented in their scale and intensity and showed that fire spalling is a common phenomenon related to wildfires, not only in semi-arid central Australia but also in the temperate environments of eastern Australia. The way we reworded the abstract delineates the aspect of flared slope development from the process of fire spalling as a weathering mechanism.

Whilst the authors correctly argue that fire-induced rock spalling “has yet to be incorporated into broad-scale landscape models”, I find the claim (L511...) “it has yet to be recognised as an important and widespread mechanism of physical weathering and sediment production” not supported by evidence. In the rebuttal it is argued that “Fire spalling is not recognised by many as an important weathering agent – more as an occasional, rare and isolated phenomena of little broader importance”. This could be seen as a matter of opinion, but the only reference supporting this claim is Nr24 (line 127), an article published in an international magazine (albeit peer-reviewed), whereas many studies also cited here have previously pointed out that fire is an important rock weathering agent, including one widely cited review article (Nr48), which highlighted even in the abstract that “Wildfire-induced weathering rates have been reported to be high relative to other weathering processes in fire-prone terrain”. In my opinion this somewhat undermines the credibility of the work presented here. The authors' statement (L128-130) “Our observations of rock surfaces following wildfires are that fire related rock spalling is a commonly observed phenomena wherever high intensity fire has swept across rocky outcrops” supports what other studies have pointed out before.

Thank you for the acknowledgement that we correctly argue that fire-induced rock spalling has yet to be incorporated into broad-scale landscape models. We acknowledge throughout that others have observed the physical effects of fire on rock weathering, going back to Blackwelder 1927 but, yes, in our excitement we see that the statement “Fire spalling is not recognised by many as an important weathering agent – more as an occasional, rare and isolated phenomena of little broader importance” might come across as over-reaching – we were trying to emphasise that there have been no real estimates of rates of erosion or sediment production specifically due to fire spalling in landscape evolution models. We have toned this down considerably and re-emphasised the early observations of fire spalling by Blackwelder in 1927 – see new text below.

“Whilst fire spalling was identified as a mechanism of physical weathering as early as 1927³² it was largely regarded as an isolated, local phenomena and thus hasn't been seriously incorporated into erosion and landscape evolution models, that currently tend to focus almost entirely on fluvial, glacial, periglacial and erosive mass wasting processes operating within drainage basin catchments.”

I do suggest to make a clearer distinction in the abstract and main text between the novel idea that (i) fire-induced rock spalling is a mechanism of physical weathering responsible for inselberg and flared slope

development and (ii) the wider relevance of fire-induced rock spalling in fire prone landscapes as a whole.

Thank-you for this suggestion. We have reworded the Abstract to make a clearer distinction between the process of flared slope development and ii) the wider relevance of fire spalling as a weathering mechanism. The two aspects are inextricably related, but we separate the two as observational fact without over-reaching by saying that fire spalling is a ubiquitous process across all landscapes – it is certainly only important in fire-prone environments and after making this observation we bring the focus back to the original aspect of flared slope development around inselbergs in semi-arid environments. We have also reduced the words to fit in with the 150 word limit – see below.

Abstract

Inselbergs, such as Uluru in central Australia, are iconic landscape features of, semi-arid and deeply denuded continental interiors. These islands of rock are commonly skirted by steep, overhanging cliffs (flared slopes) at ground level. The weathering processes responsible for formation of flared slopes and steep-sided inselbergs in flat, planated landscapes is enigmatic. One model emphasizes sub-surface weathering followed by denudation and excavation of saprolite to expose the unweathered bedrock while other models advocate slope development under subaerial conditions at ground level. We present a new hypothesis that identifies wildfire as a primary agent of flared slope development via fire-induced, rock spalling around the periphery of inselbergs. Widespread fire-spalling following the 2019-2020 Australian fires illustrates this is a common form of physical weathering in fire-prone environments but its effects are particularly evident in semi-arid regions where lateral fire-spalling dominates over fluvial and chemical weathering to create flared slopes and steep-sided inselbergs.

Specific comments:

Abstract: The statement “Fire sculpting is an important agent of geomorphic change in any fire-prone environment...but is particularly evident in hot, dry, non-glaciated and tectonically inert continental regions like Australia, where fire spalling dominates over fluvial and chemical weathering to create flared slopes and steep sided inselbergs”. is based on the authors’ hypothesis. This may well be the case, but this hypothesis has yet to be tested with empirical data and should therefore be rephrased to “Fire sculpting is an important agent of geomorphic change in any fire-prone environment...but is particularly evident in hot, dry, non-glaciated and tectonically inert continental regions like Australia, where we suggest/we hypothesise fire spalling dominates over fluvial and chemical weathering to create flared slopes and steep sided inselbergs”.

Thank you for this suggestion. We have reworded the hypothesis. Part of the abstract has been moved into the discussion. We have removed this section from the Abstract and left it to the Discussion and Conclusion following the presentation of our results and background information which as you suggest is the more correct way to present a hypothesis.

L100, Fig. 2g and elsewhere: caves (incorrectly referred to Fig. 5 in the revised manuscript). Tafoni have been removed from the manuscript given that there was no convincing evidence for a pyrogenic origin (as acknowledged in the rebuttal). The same applies to caves which share many morphological features with tafoni. Retaining them in the manuscript adds nothing tangible to the arguments about flared slopes and instead confuses the issue. Shallow caves are found in many rock types and are often morphologically similar whether they occur in fire prone and virtually fire free environments.

Yes, thank you for making this point. We have removed unnecessary reference to caves where it has any kind of genetic implications.

L141: “Average flame-front residence time for eucalypt forest fuels is about 37 seconds but radiant heat and hot winds fanning out in front of the fire have the ability to pre-heat front the rock surface and vegetation before and after arrival of the fire”. This precise statement is based on one case study using experimental fires in jarrah forest of south-west Western Australia and is by no means representative of flame residence times for wildfires in eucalyptus forests as a whole – as implied here. For example, the fire behavior model developed based on these fires underestimated fire propagation in the 2009 Black Saturday fires.

Thankyou for pointing this out. We have reworded this section to make it clear that this was a single, although very well-resourced and monitored, case study and not necessarily representative of all fires. See below. “Experimental fires lit and monitored in jarrah forests of south-west Western Australia (Project VESTA) reveal that temperature correlates directly with rate of spread, fire intensity, flame height and surface fuel bulk density²². This single case study measured average flame-front residence time in eucalypt forest fuels of about 37 seconds.”

L145 onwards: “Basalt is a high temperature volcanic rock with no quartz content. Fire spalling was minimal across most of the outcrops and generally consisted of dislodged phenocrysts. However, a few basalt outcrops adjacent nearby fallen burnt logs were intensely scorched and displayed thin (1-4 mm) spalled flakes of basalt

indicating that fire spalling is not restricted entirely to quartz-rich lithologies.” This is an important observation, which does point to a perhaps even more relevant point here and related to L141. Flame front residence times are generally short – in the order of seconds to a few minutes, EXCEPT where large woody fuels are present (termed ‘down wood’). In mature eucalypt forests, dead down wood can burn or smoulder for hours and even days, providing ideal conditions for spalling (note the remnants of these in some of your images). However, this mechanism is likely to be much more prevalent in woodlands rather than arid or semiarid regions.

Thank you. Yes, the basalt spalling was an interesting find. We specifically sought out this site to check for spalling within basaltic outcrops and it confirmed our suspicions that spalling rarely occurs in basalt. This appeared to be a particularly high-temperature fire as indicated by the glass bottle near a burnt tree trunk that melted and set without shattering – see below. We added a paragraph to address this point and noted the presence of ‘ghosted’ tree trunk impressions on some rock faces indicating some trees have burnt multiple times and spalled an impression of the trunk.

“In mature eucalypt forests with large, woody fuels, termed ‘down wood’, fires can burn or smoulder for days, providing prolonged heat required for extensive spalling. Some cliff faces record distinct ghosted impressions of nearby tree trunks with the resultant spalling hollowing out the line and shape of a tree trunk in an otherwise flat, vertical rockface that reflect that the tree has been burnt and recovered multiple times (Figure 5a – right hand side).”

See additional picture below showing ghosted impression of burnt tree trunk in the rock face.

L295: I suggest “flame height” instead of “height of the fire” here to be more precise.

Yes. Corrected.

L486-492: It is clearly useful to put the effect of spalling into the context of erosion rates as a whole. In the 2019-20 fire season, the area burned in the SE Australian forest biome was ca 10 x that for a ‘typical’ fire season (<https://www.nature.com/articles/s41558-020-0716-1>) [although the area burned in Australia as a whole was relatively low]. Hence in this biome spalling would have affected a much larger area than is normal, although this may not be very relevant when considering the long term effect of fire enhanced erosion cycles as a whole.

Yes we agree that the 2019-20 Black Summer fire season was unusually large for SE Australia but not Australia as a whole. Certainly, detailed fire-related erosion models would need to consider long term recurrence intervals which rely on paleofire studies. The number of paleofire record studies is increasing in Australia but there are still only a handful available for developing evidence based long-term erosion models in the semi-arid regions. We are not looking to provide a definitive erosion rate for any particular area in this study following any particular fire event – we are more concerned with investigating the potential rates of erosion by identifying important variables such as fire recurrence interval and rock-type. We present these variables via a working equation that has the potential to estimate long-term rates of weathering and denudation as more information comes to light. Hopefully future studies will start to quantify rates of fire-related erosion in specific biomes and we can produce more accurate landscape models that better predict long-term fire recurrence intervals, rates of erosion and sediment production. Fire spalling is an overlooked process worth consideration when interpreting rates of weathering measured by cosmogenic studies.

Rather than providing a figure for the estimated mass of spalling over the area burned in this season (100 Mill tons), it would be much more meaningful to give a figure that enables comparison with other studies that examined fire induced erosion rates in this region of SE Australia (e.g. DOI: 10.1002/esp.1460), such as t/ha or

mm/y (or mm/ky). I also doubt that the surface area of rock susceptible to spalling is equivalent to 1% of the entire land area burned. Spalling was clearly noticeable on some cliff faces for example in the Blue Mountains after the 2019/20 fires (as also depicted in Fig. 5), but this is one of the most cliff/outcrops rich areas in SE Australia. The Grose valley (Fig. 5g) has ca 20 km of cliffline in an area of 20 km². If as much as 1 m² per every m of cliffline had lost its surface from spalling in this fire, this would equate to 0.1% of the land surface affected by spalling (not 1% as used in the example by the authors), and this is likely to be one of the most cliff-dense and severely burned area within the 2019/20 area burned. In addition, 2 cm or thicker sheets of spalled rock can indeed be observed in some places burned in 2019/20 in the Blue Mountains, but this is likely to be at the high end of the spalling depth range. Thus, I suggest that the actual loss of material from rock faces in this event is likely to be orders of magnitude below that estimated (as an illustrative example) by the authors. A recalculation would bring this more in line with denudation and spalling estimates made in previous studies for this region (as summarised in DOI: 10.1002/esp.1460 and estimated by Humphreys et al. 2003. Some effects of fire on the regolith. In: Roach, I.C. (Ed.), *Advances in Regolith*. pp. 216–220.)

Thank you for the references. This is an interesting point worthy of further discussion.

Just to take up the example of the Grose Valley which contains ~20 km of cliff line in 20 km². Cliff lines are three dimensional surfaces of varying height. The base of the Grose Valley sits at 600 m elevation and the top at just over 1000 m at Blackheath and the cliff itself has a vertical height of 100-200 m – Govett's Leap Falls has a single, vertical drop of 180 m. If we take a conservative average cliff height of 100 m and use the 20 km estimate for the cliff line length, then the total area of cliff line is actually 2 km². This vertical surface area is 10% of the 20 km² area as calculated from an aerial two-dimensional perspective which is a significant observation in itself that I hadn't considered until now! Obviously, the entire cliff line isn't always spalled and most spalling will occur near the base or the top where fuel densities and flame intensities are highest. However, at Blackheath the fires raced up the cliff faces either side of Govett's Leap Falls and we observed spalling all the way up the cliff face. Spalling was particularly intense at the base and near the top where there was thick vegetation to fuel the fires but the hanging swamps perched on the cliff were burnt resulting in fire spalling on the adjacent rock faces. Thus, I think that there is much more than only 1 m² of spalled cliff face per metre of cliff line – the picture you provided previously clearly shows the vertical extent of spalling near the base of a cliff line but clearly this is dependent upon the proximity and density of flammable vegetation along the cliff line. Measuring the "length" of a cliff lines is also a bit like measuring the length of a coastline in that it is fractal in nature and the more detailed the measurement around every peninsular, gully and crevice the greater the length so the true length and therefore surface area of this cliff line is probably much larger. Cliff lines are a significant outcrop feature but not the only rocky outcrops in the catchment - there are also numerous outcrops and fallen boulders on the flanks of the valleys that will be spalled by fire as it passes through and these must be included in catchment-wide calculations of spalled rock surfaces as they are legitimately generating new sediment during fire-spalling. Without wanting to be unnecessarily contrary, it may actually be that our 1% estimate of a catchment area being affected by spalling is an underestimate of total rock surfaces spalled depending on the terrain but I think this requires much more detailed terrain and remote sensing analyses so I am happy to go with your more conservative 0.1% area spalled if you think that is more suitable. However, in terms of our estimation of sediment production following the 2019-20 fires, we wanted to give a conservative estimate of the potential amount of new sediment produced in a single but unusually large fire season such as 2019-20. We thought that 1% was a low estimate particularly across the rocky granite terranes that burnt at Cobargo and near Tamworth and we note that the Humphries 2003 reference you provided also uses 1% of land area spalled as a ball park figure – quote "*Nevertheless, if 1% of this area was so affected every 20 years a spalling rate of 3 g m² y⁻¹ results, which translates to a denudation rate of about 6 m My⁻¹ assuming a rock density of 2 g cm⁻³. This is probably a conservative estimate that does not take into account charring of the cement-weakened algal coating of exposed rock*". The percentage of area that is rocky outcrop is quite high in most of the burnt areas as the remaining bushland is generally in areas that are non-arable or so mountainous that farming and development was deemed unsuitable and consequently the land was designated into a National Park. We admit that accurately estimating the proportion of rocky outcrops potentially affected by fire spalling is difficult given the patchy, three-dimensional nature of outcrops that vary according to rocktype, along with the distribution of different vegetation types and fuel loads. It's an area for future GIS/remote sensing focussed studies. Our estimate of 1% of the areas being affected by spalling (on average 2 cm thick) is crude because no estimates of outcrop surface area or fire-spalled rock surface areas have been attempted yet. We thought 1% might be a conservative estimate just to start discussions but we are happy to tone down the estimate to 0.1% at 1cm average spall thickness which would equate to 5 million tonnes of spalled rock rather than 100 million tonnes but we emphasise that it is the concept that is important. As spatial technologies improves more precise measurements of variables such as outcropping areas, degrees of spalling and fire intensity will enhance the model accuracy in relation to sediment production rates for fire affected areas.

The second point relates to the rates of fire induced erosion of hillslope soils and sediments in the Blue Mountains has been the focus of several quantitative post-fire erosion studies as initiated by my co-authors honours project in 1994 that was later published in Dragovich and Morris (2002). However, these studies focus on the erosion of existing soil/sediment on hillslopes rather than the generation of new sediment at the rockface. The cosmogenic studies of river sediments also focus on average vertical denudation across the landscape which like the tops of inselbergs is incredibly slow. However, rates of erosion are not uniform across any landscape and fire spalling is a mechanism of physical weathering that generates new sediment at the rock-vegetation (fire) interface, particularly along steep rocky slopes or cliff lines and operates laterally rather than

vertically. Fire spalling is a completely different mechanism of denudation to post-fire movement of soil/sediment on hillslopes by water and mass movements. For this reason, we didn't think it was appropriate to compare rates of hillslope soil/sediment erosion with the rates of fire spalled weathering. We think that the apparently precise answers provided by cosmogenic studies need to be untangled more in terms of the variables that lead to the generation of new sediment and residence times on slopes and in rivers across mountainous landscapes like the Blue Mountains that are not homogenous in terms topography, slope, vegetation and rates of denudation.

We have adjusted the text to read;

'To put this in perspective, if a conservative estimate of just 0.1% of the 18.6 million hectare burnt areas across Australia in 2019-20 were rocky outcrops affected by fire-spalling, on average 1 cm thick; then this would equate to over 5 million tonnes of spalled rock during the 2019-20 fire season. If 1% of the burnt area spalled 2 cm of rock then the amount of sediment produced is closer to 100 million tonnes. Accurate estimates will vary depending upon the terrain, the outcrop, rock-type and intensity of fires of these fires were in mountainous regions with abundant rocky outcrops.'

Stefan Doerr

Reviewer #3 (Remarks to the Author):

The authors have responded to the reviewers' comments and provided additional field observations and discussion on inselberg formation. Some of the issues relating to vegetation remain somewhat unconvincing but in the absence of conclusive evidence these arguments remain subject to individual interpretation (and do not negatively impact on publication). The comments below are for authors' consideration only and do not require a formal rebuttal/reply.

The paper's title indicates that the authors consider fire-induced rock spalling is responsible for development of steep slopes on the margins of inselbergs. Are other mechanisms also involved, with the proviso that the flared slopes which appear on some inselberg margins may be distinctively and directly related to fire? Perhaps a word change in the title may be appropriate – Fire-induced rock spalling as a mechanism of physical weathering responsible for flared slope development on inselbergs (??)

Yes. Thank you! We like the adjustment to the title as it logically starts with flared slope development and leads into the bigger picture of steep sided inselberg formation. We have also reduced the number of characters to fit with requirements and will go with -

Fire-induced rock spalling as a mechanism of weathering responsible for flared slope and inselbergs development

It is implied that all rock outcrops are composed of relatively fresh rock. Equation 1 may benefit from a clear statement of this assumption, especially as the application of the model includes both wet temperate and hot desert environments – the factor of rock strength as well as rock type is relevant to W . In wetter climates, sandstone surfaces which are potentially more weathered, of lower strength and with a thicker weathering 'rind' may respond by disintegrating in a high intensity fire rather than by spalling. Regardless of the process, fire may lead to similar morphological outcomes on the outcrop but potentially different rates of surface change over time. Consider ' $W = \text{Average} \dots \text{Dependent on rock type (quartz content), rock strength, fire} \dots$ '

OK thank you. Yes, there is much experimental work yet to be done on how different rock types with varying strengths, compositions, degrees of chemical weathering etc react to fire. We added "rock strength" as suggested. Texture is also something to consider – we noticed that conglomerates seem to be less susceptible to spalling than fine sandstones but again this requires further detailed experimental testing on different rock types.

Very minor editing – Add reference 34 in paragraph commencing 'Although fire was first identified....back in 1927 34 it has yet...'

Thankyou. Reference added

REVIEWERS' COMMENTS

Reviewer #2 (Remarks to the Author):

The authors have very thoroughly and convincingly addressed all remaining comments. This is now a well argued and fascinating study that I hope will be widely read. I recommend acceptance as is.

(A minor point for the typesetter: Reference 36 should read 'Zeitschrift für' instead of 'Zeitschriffiir')